# Discrete Modeling via Boundary Conditional Diffusion Processes

**Yuxuan Gu**[†]  **Xiaocheng Feng**[†‡]  **Lei Huang**[†]  **Yingsheng Wu**[†]  **Zekun Zhou**[†]
**Weihong Zhong**[†]  **Kun Zhu**[†]  **Bing Qin**[†‡]

[†]Harbin Institute of Technology  [‡] Peng Cheng Laboratory

{yxgu,xcfeng,lhuang,yswu,zkzhou,whzhong,kzhu,qinb}@ir.hit.edu.cn

## Abstract

We present an novel framework for efficiently and effectively extending the powerful continuous diffusion processes to discrete modeling. Previous approaches have suffered from the discrepancy between discrete data and continuous modeling. Our study reveals that the absence of guidance from discrete boundaries in learning probability contours is one of the main reasons. To address this issue, we propose a two-step forward process that first estimates the boundary as a prior distribution and then rescales the forward trajectory to construct a boundary conditional diffusion model. The reverse process is proportionally adjusted to guarantee that the learned contours yield more precise discrete data. Experimental results indicate that our approach achieves strong performance in both language modeling and discrete image generation tasks. In language modeling, our approach surpasses previous state-of-the-art continuous diffusion language models in three translation tasks and a summarization task, while also demonstrating competitive performance compared to auto-regressive transformers. Moreover, our method achieves comparable results to continuous diffusion models when using discrete ordinal pixels and establishes a new state-of-the-art for categorical image generation on the CIFAR-10 dataset.

## 1  Introduction

Discrete modeling is essential due to the natural prevalence of discreteness in numerous domains, including proteins [Madani et al., 2020, 2023], images [Parmar et al., 2018, Dosovitskiy et al., 2021], and natural language [Sutskever et al., 2014, Brown et al., 2020]. Recent dominant framework for discrete modeling is the Transformer [Vaswani et al., 2017] with an autoregressive manner. While achieving impressive performance, it does suffer from a slow step-by-step generation process, especially for long sequences. Continuous Diffusion models [Sohl-Dickstein et al., 2015, Ho et al., 2020], on the contrary, exhibit the ability to recover high-dimensional data from noise in parallel with limited iteration steps. Although proved to be effective in continuous data generation [Rombach et al., 2022, Kong et al., 2021], they continue to encounter challenges in discrete modeling [Austin et al., 2021, Chen et al., 2023b, Li et al., 2022, Gong et al., 2023b].

In this paper, we reveal a significant discrepancy pertaining to the modeling of discrete data using continuous diffusion models. Current approaches represent a discrete sample with a vector point in the continuous space. The diffusion process learns a neural network to model the probability distributions that recovers this continuous point from Gaussian noise. However, the discrete data actually corresponds to an area in the continuous space rather than a single point, where the oversimplified assumption leads to a mismatch between learned probability contours and the boundary of the discrete area. Take language generation as an example, a word is represented with an embedding vector in the embedding space. To generate this word, it is impractical to strictly enforce the predicted vector to be an exact match to the embedding. On the contrary, vectors around this embedding can also generate

the same word, thereby defining the collective area they encompass as a discrete area of this word. As illustrated in Figure 1A, suppose the learned probability density function is $p_\theta(\mathbf{x})$ and two points $\mathbf{x}^i$ and $\mathbf{x}^o$ are sampled in the same density contour where $p_\theta(\mathbf{x}^i) = p_\theta(\mathbf{x}^o)$. It is obvious that $\mathbf{x}^i$ lies in the discrete area and is able to recover the discrete data while $\mathbf{x}^o$ can not. This means that the diffusion model only learns a simplified scenario that does not match the real probability distribution.

To address the issues above, we proposed to take the boundaries of discrete areas as priors, as shown in Figure 1B, where boundary curves are regarded as oracle contours. As it gradually approaches the discrete boundary, the learned density contours of diffusion models are expected to transform from Gaussian distributions to the boundary distribution. Therefore, we propose to divide the forward process into two steps. First is the boundary estimation where we precisely calculate the stopping time $t_0$ and position $\mathbf{x}_{t_0}$ at which the forward trajectory cross the boundary. Then we rescale the trajectory for both training and inference stages to make the sampling probability of noisy point $\mathbf{x}_t$ conditioned on the boundary. To make the boundary estimation tractable (appendix A) and eliminate randomness in conditional state transitions $\mathbf{x}_{t_0} \rightarrow \mathbf{x}_t$, we utilize the Ordinary Differential Equations (ODEs) to describe the forward trajectory.

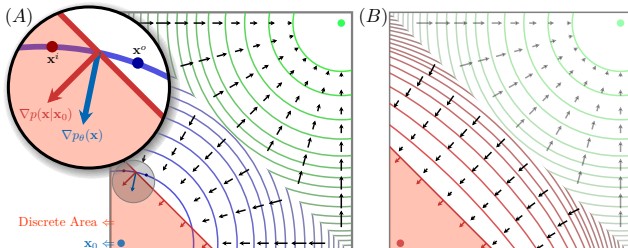

Figure 1: (A) Blue and green curves are the learned probability density contours of the diffusion model for two data points. The red area is the discrete area of the blue data $\mathbf{x}_0$ and the boundary of this area is naturally a density contour. The discrete boundary is a complex hypersurface in the high-dimensional continuous space and we simplify it into a red line for convenience of description. As observed in the magnified part, the learned contours deviate from the boundary contour, resulting in inconsistent probability densities and gradient directions. (B) We consider the discrete boundary as priors for the diffusion process to estimate a more appropriate probability distribution, where the learned contours are expected to follow the shape of the discrete boundary.

Our approach is experimented in both language modeling and discrete image generation. On three machine translation datasets (IWSLT14 DE-EN [Cettolo et al., 2012], WMT14 EN-DE, WMT16 EN-RO) and a text summarization dataset (GIGAWORD [Rush et al., 2015]) for language modeling, our proposed approach not only significantly improves existing diffusion models to at most 7.8% but also achieves competitive performance to autoregressive transformers. For image generation on CIFAR-10 [Krizhevsky et al., 2009], our model realizes a comparable result to continuous diffusion models with discrete ordinal pixels and establishes a new state-of-the-art for categorical pixels.

## 2 Preliminaries

**Diffusion Models** To model a real distribution $q(\mathbf{x}_0)$, diffusion models utilize a forward process $p_t(\mathbf{x}|\mathbf{x}_0)$ with $T$ steps to gradually add Gaussian noise $\pi(\mathbf{x}) = \mathcal{N}(\mathbf{0}, \mathbf{I})$ into the data distribution, where $p_T(\mathbf{x}|\mathbf{x}_0) = \pi(\mathbf{x})$. There are different architectures for the forward process. A common approach [Ho et al., 2020] considers the forward process as the Markovian process, where $p_t(\mathbf{x}|\mathbf{x}_0) = \prod_{s=1}^{t} p_s(\mathbf{x}_s|\mathbf{x}_{s-1})$ combines a series of Gaussian distributions. Thus the forward process follows a Gaussian distribution that $p_t(\mathbf{x}|\mathbf{x}_0) = \mathcal{N}(\sqrt{\bar{\alpha}_t}\mathbf{x}_0, (1 - \bar{\alpha}_t)\mathbf{I})$ (Variance Preserving) or $p_t(\mathbf{x}|\mathbf{x}_0) = \mathcal{N}(\mathbf{x}_0, \sigma_t^2\mathbf{I})$ (Variance Exploding) [Song et al., 2021b], where noise scheduler $\bar{\alpha}_t$ monotonically decreases from 1 to 0 and $\sigma_t$ increases from sufficiently small to the maximum pairwise distance between all training data points. To recover data from noise, diffusion processes train neural networks $\mathbf{x}_\theta(\mathbf{x}_t, t)$ to predict $\mathbf{x}_0$ (other equivalent targets include $\boldsymbol{\epsilon}$ and $\nabla \log p(\mathbf{x}_t)$) from $\mathbf{x}_t \sim p_t(\mathbf{x}|\mathbf{x}_0)$:

$$\mathcal{L}_\theta = \mathbb{E}_{t \sim \mathcal{U}_{(1,T)}, \mathbf{x}_0 \sim q(\mathbf{x}_0), \mathbf{x}_t \sim p_t(\mathbf{x}|\mathbf{x}_0)} \left[ \|\mathbf{x}_0 - \mathbf{x}_\theta(\mathbf{x}_t, t)\|^2 \right]. \tag{1}$$

Samples are generated with a series of reverse state transition $p(\mathbf{x}_{t-1}|\mathbf{x}_t, \mathbf{x}_\theta(\mathbf{x}_t, t))$.

**Flow Matching** Another architecture [Lipman et al., 2023] utilizes the ODEs and defines a time-dependent flow function $\phi_t(\mathbf{x}) = \sigma_t(\mathbf{x}_0)\mathbf{x} + \mu_t(\mathbf{x}_0)$ that maps $p_t(\mathbf{x}|\mathbf{x}_0) = [\phi_t]_* \pi(\mathbf{x}) = \pi(\phi_t^{-1}(\mathbf{x})) \left| \det \frac{\mathrm{d}\phi_t^{-1}(\mathbf{x})}{\mathrm{d}\mathbf{x}} \right| = \mathcal{N}(\mu_t(\mathbf{x}_0), \sigma_t^2(\mathbf{x}_0)\mathbf{I})$, where $\mu_t$ and $\sigma_t$ can be the same as in diffusion

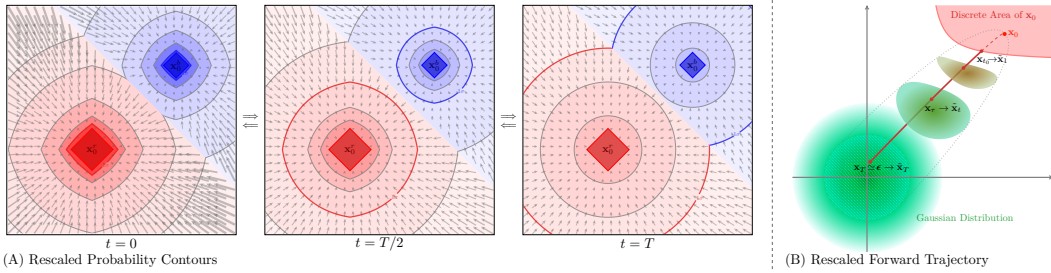

| | | | |
|---|---|---|---|
| $t = 0$ | $t = T/2$ | $t = T$ | |
| (A) Rescaled Probability Contours | | | (B) Rescaled Forward Trajectory |

Figure 2: (A) Rescaled Probability Contours. The bold curve $1\sigma$ is the density contour of one standard deviation. As the time $t$ decreases from $T$ to 0, the rescaled contours will gradually fit the discrete boundary and probability densities will also concentrate to this boundary. (B) Rescaled Forward Trajectory. Original forward trajectory $\mathbf{x}_0 \rightarrow \mathbf{x}_{t_0} \rightarrow \mathbf{x}_\tau$ is rescaled to be a boundary conditional trajectory $\tilde{\mathbf{x}}_1 \rightarrow \tilde{\mathbf{x}}_t$ that starts from $\tilde{\mathbf{x}}_1 = \mathbf{x}_{t_0}$. The rescaled forward distribution $\tilde{p}_t(\tilde{\mathbf{x}}_t|\mathbf{x}_0)$ is transformed from the discrete boundary to Gaussian distributions.

models or a more straightforward form that $\mu_t = (1 - \frac{t}{T})\mathbf{x}_0$ and $\sigma_t = \frac{t}{T}$. Recovering data from noises relies on the vector field $u_t(\mathbf{x}|\mathbf{x}_0)$ that generates the probability path with the ODE $d\phi_{T-t}(\mathbf{x}) = u_{T-t}(\phi_{T-t}(\mathbf{x})|\mathbf{x}_0)dt, t : 0 \rightarrow T$. Neural networks $u_\theta(\mathbf{x}, t)$ are trained to estimate the vector field $u_t(\mathbf{x}|\mathbf{x}_0)$ via the following objective:

$$\mathcal{L}_\theta = \mathbb{E}_{t \sim \mathcal{U}_{(1,T)}, \mathbf{x}_0 \sim q(\mathbf{x}_0), \mathbf{x}_T \sim \pi(\mathbf{x})} \left[ \left\| u_\theta(\phi_t(\mathbf{x}_T), t) - \frac{d\phi_t(\mathbf{x}_T)}{dt} \right\|^2 \right]. \tag{2}$$

Besides, the vector field is proved to have the form:

$$u_t(\mathbf{x}|\mathbf{x}_0) = \frac{\sigma'_t(\mathbf{x}_0)}{\sigma_t(\mathbf{x}_0)} (\mathbf{x} - \mu_t(\mathbf{x}_0)) + \mu'_t(\mathbf{x}_0), \text{ where apostrophe indicates derivative to } t. \tag{3}$$

## 3 Methodology

As illustrated in Figure 2, our objective is to refine the probability density contours of $p_t(\mathbf{x}|\mathbf{x}_0)$ so that they better fit the boundaries of discrete samples while still allowing for the ease of sampling. Let $\mathbf{x}_0$ denote the samples from a real distribution $q(\mathbf{x}_0)$. Obtaining a boundary-aware corresponding noisy data $\mathbf{x}$ at time $t \in [1, T]$ is $p_t(\mathbf{x}|\mathbf{x}_0) = \int p_t(\mathbf{x}, \mathbf{x}_{t_0}, t_0|\mathbf{x}_0)d\mathbf{x}_{t_0}dt_0$, where $t_0$ is a random variable distributed according to when the diffusion trajectory and the discrete boundary intersect, and $\mathbf{x}_{t_0}$ is the corresponding sample point at $t_0$. Then the forward process is rescaled in two steps:

$$\tilde{p}_t(\mathbf{x}|\mathbf{x}_0) = \int \underbrace{\tilde{p}_t(\mathbf{x}|\mathbf{x}_{t_0}, t_0, \mathbf{x}_0)}_{\text{Trajectory Rescaling}} \underbrace{p(\mathbf{x}_{t_0}, t_0|\mathbf{x}_0)}_{\text{Boundary Estimation}} d\mathbf{x}_{t_0}dt_0, \tag{4}$$

where the latter term is to calculate the discrete boundaries and the former term is to rescale the forward trajectory. In order to make the equation tractable and ensure that $\mathbf{x}$ and $\mathbf{x}_{t_0}$ are on the same trajectory, we model the forward process with flow functions $\phi_t(\mathbf{x})$ and extend the notation as:

$$\psi_t(\mathbf{x}) = \mathbf{u}(\mathbf{x}_0, t)\,\mathbf{x}_0 + \mathbf{v}(\mathbf{x}_0, t)\,\mathbf{x}, \quad p_t(\mathbf{x}|\mathbf{x}_0) = [\psi_t]_*\pi(\mathbf{x}) \tag{5}$$

where $\mathbf{u}(\cdot)$ and $\mathbf{v}(\cdot)$ are coefficient functions and sampling $\mathbf{x}_t$ from $p_t(\mathbf{x}|\mathbf{x}_0)$ equals to

$$\mathbf{x}_t = \psi_t(\boldsymbol{\epsilon}), \quad \boldsymbol{\epsilon} \sim \pi(\mathbf{x}) = \mathcal{N}(\mathbf{0}, \mathbf{I}). \tag{6}$$

### 3.1 Estimate Discrete Boundaries

Before figuring out the joint distribution $p(\mathbf{x}_{t_0}, t_0|\mathbf{x}_0)$, let's start by discussing how to verify whether an arbitrary point $\mathbf{x}$ in the continuous space belongs to the discrete area of $\mathbf{x}_0$. Suppose $\mathbf{x}_0$, which exists in the continuous space $S$, is the representation vector of a discrete random variable $\mathcal{I}$ in a discrete space with $K$ states. Besides, $\mathcal{J}$ is another discrete random variable *i.i.d.* with $\mathcal{I}$. We define the discrete area of $\mathbf{x}_0$ in the continuous space $S$ as:

$$C_\mathcal{I} = \{\forall \mathbf{x} \in S | f(\mathbf{x}, \mathcal{I}) > f(\mathbf{x}, \mathcal{J}), \forall \mathcal{J} \neq \mathcal{I}\}, \tag{7}$$

where $f(\mathbf{x}, \mathcal{I})$ is a function assessing the likelihood of an arbitrary continuous point $\mathbf{x}$ inside the discrete area of $\mathbf{x}_0$. For instance, in language modeling, $K$ is the vocabulary size. $\mathcal{I}, \mathcal{J} \in K^n$ are two different sequences of $n$ tokens and $\mathbf{x}_0 \in \mathbb{R}^{[n,m]}$ is a sequence of $m$-dimensional vector embeddings for $\mathcal{I}$. $f(\mathbf{x}, \mathcal{I})$ is the dot similarity function. $C_{\mathcal{I}}$ collects all vectors in the embedding space that will be decoded to generate $\mathcal{I}$ and excludes vectors associated with any other token sequences $\mathcal{J}$.

Given a noisy point $\mathbf{x}_{t_0}$ locating at the boundary between $C_{\mathcal{I}}$ and $C_{\mathcal{J}}$, we can get $|f(\mathbf{x}_{t_0}, \mathcal{I}) - f(\mathbf{x}_{t_0}, \mathcal{J})| = 0$ based on previous definition. Replacing $\mathbf{x}_{t_0}$ with eqs. (5) and (6), there is:

$$f(\mathbf{u}_{t_0}\mathbf{x}_0 + \mathbf{v}_{t_0}\boldsymbol{\epsilon}, \mathcal{I}) = f(\mathbf{u}_{t_0}\mathbf{x}_0 + \mathbf{v}_{t_0}\boldsymbol{\epsilon}, \mathcal{J}). \tag{8}$$

In language modeling and categorical images, $f(\cdot)$ is a linear projection function that:

$$\mathbf{u}_{t_0}(f(\mathbf{x}_0, \mathcal{I}) - f(\mathbf{x}_0, \mathcal{J})) = \mathbf{v}_{t_0}(f(\boldsymbol{\epsilon}, \mathcal{J}) - f(\boldsymbol{\epsilon}, \mathcal{I})). \tag{9}$$

Further simplification of this equation can not be universally applied to all arbitrary forms of $\mathbf{u}_{t_0}$ and $\mathbf{v}_{t_0}$. Therefore, we calculate separately for several commonly occurring special cases.

**Diffusion Process**  For variance preserving, there is $\mathbf{u}_t^2 + \mathbf{v}_t^2 = 1$ and we have:

$$\mathbf{u}_{t_0} = 1 \bigg/ \sqrt{1 + \left(\frac{f(\mathbf{x}_0, \mathcal{I}) - f(\mathbf{x}_0, \mathcal{J})}{f(\boldsymbol{\epsilon}, \mathcal{J}) - f(\boldsymbol{\epsilon}, \mathcal{I})}\right)^2} \text{ and } \mathbf{v}_{t_0} = 1 \bigg/ \sqrt{1 + \left(\frac{f(\boldsymbol{\epsilon}, \mathcal{J}) - f(\boldsymbol{\epsilon}, \mathcal{I})}{f(\mathbf{x}_0, \mathcal{I}) - f(\mathbf{x}_0, \mathcal{J})}\right)^2}.$$
$$\tag{10}$$

For variance exploding, there are $\mathbf{u}_t = 1$ and $\mathbf{v}_t = \sigma_t$. We can obtain:

$$\mathbf{u}_{t_0} = 1 \text{ and } \mathbf{v}_{t_0} = (f(\boldsymbol{\epsilon}, \mathcal{J}) - f(\boldsymbol{\epsilon}, \mathcal{I})) / (f(\mathbf{x}_0, \mathcal{I}) - f(\mathbf{x}_0, \mathcal{J})). \tag{11}$$

**Flow Matching**  For optimal transport, there is $\mathbf{u}_t + \mathbf{v}_t = 1$ and similarly we get:

$$\mathbf{u}_{t_0} = 1 \bigg/ \left(1 + \frac{f(\mathbf{x}_0, \mathcal{I}) - f(\mathbf{x}_0, \mathcal{J})}{f(\boldsymbol{\epsilon}, \mathcal{J}) - f(\boldsymbol{\epsilon}, \mathcal{I})}\right) \text{ and } \mathbf{v}_{t_0} = 1 \bigg/ \left(1 + \frac{f(\boldsymbol{\epsilon}, \mathcal{J}) - f(\boldsymbol{\epsilon}, \mathcal{I})}{f(\mathbf{x}_0, \mathcal{I}) - f(\mathbf{x}_0, \mathcal{J})}\right). \tag{12}$$

As a result, $t_0$ can be directly derived by inverting the coefficient function $\mathbf{u}_t$ or $\mathbf{v}_t$, which depends on the choice of noise scheduling strategies. Since their differences do not affect our results, we omit the detailed calculation (appendix E) and denote this process with a function $G(\cdot)$:

$$t_0 = G(\mathbf{x}_0, \boldsymbol{\epsilon}), \text{ where } \mathbf{u}(\mathbf{x}_0, G(\mathbf{x}_0, \boldsymbol{\epsilon})) = \mathbf{u}_{t_0} \text{ and } \mathbf{v}(\mathbf{x}_0, G(\mathbf{x}_0, \boldsymbol{\epsilon})) = \mathbf{v}_{t_0}. \tag{13}$$

It's worth noting that $t_0$ is not a scalar but a vector, where the dimension is the number of elements in $\mathbf{x}_0$. If $\mathbf{x}_0$ is a sequence of $n$ tokens, $t_0 \in [1, T]^n$. If $\mathbf{x}_0$ is a RGB image with 3-channel $\times$ $h$-height $\times$ $w$-width of pixels, $t_0 \in [1, T]^{3 \times h \times w}$. Furthermore, the corresponding noisy sample $\mathbf{x}_{t_0}$ is derived as:

$$\mathbf{x}_{t_0} = \mathbf{u}(\mathbf{x}_0, G(\mathbf{x}_0, \boldsymbol{\epsilon}))\mathbf{x}_0 + \mathbf{v}(\mathbf{x}_0, G(\mathbf{x}_0, \boldsymbol{\epsilon}))\boldsymbol{\epsilon} = \psi_{G(\mathbf{x}_0, \boldsymbol{\epsilon})}(\boldsymbol{\epsilon}), \tag{14}$$

which is a time-independent function of the Gaussian noise $\boldsymbol{\epsilon}$. It's worth mentioning that both $p(t_0|\mathbf{x}_0)$ and $p(\mathbf{x}_{t_0}|\mathbf{x}_0)$ are intractable, since $G(\mathbf{x}_0, \boldsymbol{\epsilon})$ and $\psi_{G(\mathbf{x}_0, \boldsymbol{\epsilon})}(\boldsymbol{\epsilon})$ are not invertible to $\boldsymbol{\epsilon}$. Different $\boldsymbol{\epsilon}$s can be mapped to a same $t_0$ or $\mathbf{x}_{t_0}$. Fortunately, there is an one-to-one mapping between $\boldsymbol{\epsilon}$ and the $[\mathbf{x}_{t_0}; t_0]$ pair. We denote the boundary flow function and the corresponding inversion as

$$\Psi(\boldsymbol{\epsilon}) = [\psi_{G(\mathbf{x}_0, \boldsymbol{\epsilon})}(\boldsymbol{\epsilon}); G(\mathbf{x}_0, \boldsymbol{\epsilon})], \qquad \Psi^{-1}([\mathbf{x}_{t_0}; t_0]) = (\mathbf{x}_{t_0} - \mathbf{u}(\mathbf{x}_0, t_0)\mathbf{x}_0)/\mathbf{v}(\mathbf{x}_0, t_0), \tag{15}$$

and the joint boundary distribution is calculated as

$$p(\mathbf{x}_{t_0}, t_0|\mathbf{x}_0) = [\Psi]_* \pi([\mathbf{x}_{t_0}; t_0]). \tag{16}$$

The support set of $\mathbf{x}_{t_0}$ is restricted to the boundary contour, while other regions in the space are assigned a probability of $0$. To obtain the complete boundary, it is necessary to iterate over all possible choices of $\mathcal{J}$ and perform pairwise comparisons with $\mathcal{I}$. The complexity is $O(n \times K)$, where $n$ elements in $\mathbf{x}_0$ is independently iterated. In practical implementation, obtaining the tightest boundary only requires one step of parallel calculation and an extra $\min(\cdot)$ function over all $t_0$ candidates.

**Confidence Factor**   The discrete area defined by eq. (7) represents an ideal scenario in which the confidence of the boundary is insufficiently reliable for practical application. Due to the intractability of obtaining the probability density function across the entire discrete area and calculating its confidence interval, we employ an empirical strategy. This approach involves utilizing a confidence factor, denoted as $r$, ranging from 0 to 1, which is multiplied by $t_0$ to strike a balance between confidence and discreteness. Therefore, $r = 0$ implies the exclusion of discrete priors, causing the discrete area to collapse into a single point, which is the original diffusion process. As the value of $r$ increases, the modeling of discrete boundaries improves at the expense of reliability. Empirically, when the model is conditioned with good guidance, setting a larger value for $r$ allows us to obtain better discrete priors. However, in the case of unconditional modeling, maintaining reliability becomes more crucial to prevent oscillations and even collapses during training.

## 3.2   Rescale the Forward Trajectory

In this section, we introduce how to formulate the forward trajectory conditioned on discrete boundaries and derive the rescaled noisy sampling distribution. We start with the boundary-independent forward process $p_t(\mathbf{x}|\mathbf{x}_0)$. Let $\mathbf{x}_t$ denote a noisy point at time $t$ sampled from $p_t(\mathbf{x}|\mathbf{x}_0)$, there is $\epsilon_t = (\mathbf{x}_t - \mathbf{u}(\mathbf{x}_0, t)\mathbf{x}_0)/\mathbf{v}(\mathbf{x}_0, t)$ given eq. (5). Equations (13) and (14) provide the corresponding $[\mathbf{x}_{t_0}; t_0]$ pair on the same trajectory, which is deterministically calculated with no randomness:

$$[\mathbf{x}_{t_0}; t_0] = \Psi(\epsilon_t), \quad \text{where } \epsilon_t = (\mathbf{x}_t - \mathbf{u}(\mathbf{x}_0, t)\mathbf{x}_0)/\mathbf{v}(\mathbf{x}_0, t). \tag{17}$$

To model the transition probability $p_t(\mathbf{x}_{t_0}, t_0|\mathbf{x}_t, \mathbf{x}_0)$, we utilize the Dirac delta function $\delta(\mathbf{x}) \simeq \lim_{\sigma \to 0} \mathcal{N}(\mathbf{0}, \sigma^2 \mathbf{I})$, which can be loosely thought of as aggregating all probability densities toward the origin, assigning an infinite density at the origin and zero densities elsewhere. Therefore, we have $p_t(\mathbf{x}_{t_0}, t_0|\mathbf{x}_t, \mathbf{x}_0) = \delta([\mathbf{x}_{t_0}; t_0] - \Psi(\epsilon_t))$. Then the forward process, conditioned on the discrete boundary, is simply derived via Bayes' rule:

$$p_t(\mathbf{x}_t|\mathbf{x}_{t_0}, t_0, \mathbf{x}_0) = p_t(\mathbf{x}_{t_0}, t_0|\mathbf{x}_t, \mathbf{x}_0)\frac{p_t(\mathbf{x}_t|\mathbf{x}_0)}{p(\mathbf{x}_{t_0}, t_0|\mathbf{x}_0)} = \begin{cases} 0, & [\mathbf{x}_{t_0}; t_0] \neq \Psi(\epsilon_t) \\ +\infty \times \dfrac{p_t(\mathbf{x}_t|\mathbf{x}_0)}{p(\mathbf{x}_{t_0}, t_0|\mathbf{x}_0)}, & \text{otherwise} \end{cases}. \tag{18}$$

Since $p_t(\mathbf{x}_t|\mathbf{x}_0) > 0$ and $p(\mathbf{x}_{t_0}, t_0|\mathbf{x}_0) > 0$, $p_t(\mathbf{x}_t|\mathbf{x}_{t_0}, t_0, \mathbf{x}_0)$ is also a delta function that

$$p_t(\mathbf{x}_t|\mathbf{x}_{t_0}, t_0, \mathbf{x}_0) = \delta\left(\mathbf{x}_t - \mathbf{u}(\mathbf{x}_0, t)\mathbf{x}_0 - \mathbf{v}(\mathbf{x}_0, t)\Psi^{-1}([\mathbf{x}_{t_0}; t_0])\right). \tag{19}$$

Based on the translation property of the Dirac delta function, i.e. $\int f(x)\delta(x - a)\mathrm{d}x = f(a)$, the original forward process $p_t(\mathbf{x}_t|\mathbf{x}_0) = [\psi_t \circ \Psi^{-1} \circ \Psi]_* \pi(\mathbf{x}_t) = [\psi_t]_* \pi(\mathbf{x}_t)$ naturally ignores the influence of discrete boundaries, even if the boundary information is explicitly added as a condition.

To enable the discrete priors, we propose a simple and intuitive approach: rescale the forward trajectory. As shown in Figure 2B, the original forward process flows from $\mathbf{x}_0$ to a random noise $\epsilon$, and we reset the starting point to $\mathbf{x}_{t_0}$. Accordingly, the intermediate noisy points $\mathbf{x}_t, t \in [1, T]$ will be proportionally mapped on this new path, which is

$$\tilde{\mathbf{x}}_t = \mathbf{x}_\tau, \quad \tau = \mathcal{T}(t, t_0) = r \times t_0 + t \times (T - r \times t_0)/T$$
$$= \mathbf{u}(\mathbf{x}_0, \mathcal{T}(t, t_0))\mathbf{x}_0 + \mathbf{v}(\mathbf{x}_0, \mathcal{T}(t, t_0))\Psi^{-1}([\mathbf{x}_{t_0}; t_0]). \tag{20}$$

Similar to eq. (19), the rescaled conditional forward process is a Dirac delta function:

$$\tilde{p}_t(\tilde{\mathbf{x}}_t|\mathbf{x}_{t_0}, t_0, \mathbf{x}_0) = \delta\left(\tilde{\mathbf{x}}_t - \mathbf{u}(\mathbf{x}_0, \mathcal{T}(t, t_0))\mathbf{x}_0 - \mathbf{v}(\mathbf{x}_0, \mathcal{T}(t, t_0))\Psi^{-1}([\mathbf{x}_{t_0}; t_0])\right). \tag{21}$$

However, $\tilde{p}_t(\tilde{\mathbf{x}}_t|\mathbf{x}_0)$ faces the same problem of irreversibility as in eq. (14) and we derive it as:

$$\tilde{p}_t(\tilde{\mathbf{x}}_t|\mathbf{x}_0) = \int \tilde{p}_t(\tilde{\mathbf{x}}_t, \tau|\mathbf{x}_0)\mathrm{d}\tau = \int \tilde{p}_t(\tilde{\mathbf{x}}_t, \tau|\mathbf{x}_{t_0}, t_0, \mathbf{x}_0)p(\mathbf{x}_{t_0}, t_0|\mathbf{x}_0)\mathrm{d}[\mathbf{x}_{t_0}; t_0]\mathrm{d}\tau$$
$$= \int [\psi_\tau \circ \Psi^{-1} \circ \Psi]_* \pi([\tilde{\mathbf{x}}_t; \tau])\mathrm{d}\tau = \int [\psi_\tau]_* \pi([\tilde{\mathbf{x}}_t; \tau])\mathrm{d}\tau. \tag{22}$$

Obtaining the probability density function requires gathering together the probability densities of the same location $\tilde{\mathbf{x}}_t$ with different $\tau$, which is intractable. Fortunately, we only need to sample noiy points from this probability distribution $\tilde{\mathbf{x}}_t \sim \tilde{p}_t(\tilde{\mathbf{x}}_t|\mathbf{x}_0)$, which is easy to implement:

$$\tilde{\mathbf{x}}_t = \mathbf{u}(\mathbf{x}_0, \mathcal{T}(t, G(\mathbf{x}_0, \epsilon)))\mathbf{x}_0 + \mathbf{v}(\mathbf{x}_0, \mathcal{T}(t, G(\mathbf{x}_0, \epsilon)))\epsilon, \quad \epsilon \sim \pi(\mathbf{x}). \tag{23}$$

## 3.3   Recover Data from Noise

**Training Objective**    Theoretically, the diffusion neural networks can be trained as in eq. (2), where the rescaled vector field is derived as $\tilde{u}_t = \frac{d\tilde{\mathbf{x}}_t}{dt} = \frac{d\tilde{\mathbf{x}}_t}{d\tau}\frac{d\tau}{dt}$. However, since a low error estimation on $\mathbf{x}_0$ is of significant importance to our trajectory rescaling method, according to eqs. (10) to (13), we convert the objective to an upper bound of the eq. (2) (See appendix F for more details) and train a neural network $\mathbf{x}_\theta(\tilde{\mathbf{x}}_t, t)$ to predict $\mathbf{x}_0$ directly:

---

**Algorithm 1** Training

1: **repeat**
2:     $\mathbf{x}_0 \sim q(\mathbf{x}_0), \boldsymbol{\epsilon} \sim \pi(\mathbf{x}) = \mathcal{N}(\mathbf{0}, \mathbf{I})$
3:     $t \sim \text{Uniform}(\{1, \ldots, T\})$
4:     $\tau := \mathcal{T}(t, G(\mathbf{x}_0, \boldsymbol{\epsilon}))$    // eqs. (13) and (20)
5:     $\hat{\mathbf{x}}_t := \mathbf{u}(\mathbf{x}_0, \tau)\mathbf{x}_0 + \mathbf{v}(\mathbf{x}_0, \tau)\boldsymbol{\epsilon}$    // eq. (23)
6:     Take gradient descent step on
         $\nabla_\theta \|\mathbf{x}_0 - \mathbf{x}_\theta(\tilde{\mathbf{x}}_t, t)\|^2$    // eq. (24)
7: **until** converged

---

$$\mathcal{L}_\theta = \mathbb{E}_{\mathbf{x}_0 \sim q(\mathbf{x}_0), t \sim \mathcal{U}_{(1,T)}, \tilde{\mathbf{x}}_t \sim \tilde{p}_t(\mathbf{x}|\mathbf{x}_0)} \left[\|\mathbf{x}_0 - \mathbf{x}_\theta(\tilde{\mathbf{x}}_t, t)\|^2\right]. \tag{24}$$

The training procedure is demonstrated in algorithm 1 and key steps are summarized in the line 4.

**Reverse Process**    A direct approach that follows the flow matching is to solve the ODE of $d\psi_{T-t}(\mathbf{x}) = \tilde{u}_{T-t}(\psi_{T-t}(\mathbf{x})|\mathbf{x}_0)dt, \psi_T(\mathbf{x}) \sim \pi(\mathbf{x})$. This form of transformation is inefficient with $\mathbf{x}_0$-prediction during inference because we have to solve the equation of $\tau = \mathcal{T}\left(t, G\left(\mathbf{x}_\theta, \frac{\tilde{\mathbf{x}}_t - \mathbf{u}(\mathbf{x}_\theta, \tau)\mathbf{x}_\theta}{\mathbf{v}(\mathbf{x}_\theta, \tau)}\right)\right)$ to get the $\tau$ with respect to the change of $\tilde{\mathbf{x}}_t$ and $\mathbf{x}_\theta$ in real time. Therefore, we provide a deterministic reverse process as an alternative, which is a special case of DDIM [Song et al., 2021a] or the ODE with discrete timesteps. Given the time intervals $\Delta t \in [\Delta t_1, \ldots \Delta t_s], \sum \Delta t = T$, we general-

---

**Algorithm 2** Sampling

1: $t := T, \tau := T$
2: $\hat{\boldsymbol{\epsilon}} \simeq \tilde{\mathbf{x}}_t \sim \mathcal{N}(\mathbf{0}, \mathbf{I})$    // Initialing
3: **for** $\Delta t := \Delta t_1, \ldots, \Delta t_s$ **do**    // $\sum_{\Delta t} = T$
4:     $\hat{\mathbf{x}}_0 := \mathbf{x}_\theta(\tilde{\mathbf{x}}_t, t)$    // Pseudo Target
5:     $t := t - \Delta t$    // Updating
6:     $\tau := \mathcal{T}(t, G(\hat{\mathbf{x}}_0, \hat{\boldsymbol{\epsilon}}))$    // eq. (25)
7:     $\tilde{\mathbf{x}}_t := \mathbf{u}(\hat{\mathbf{x}}_0, \tau)\hat{\mathbf{x}}_0 + \mathbf{v}(\hat{\mathbf{x}}_0, \tau)\hat{\boldsymbol{\epsilon}}$
8:     $\hat{\boldsymbol{\epsilon}} := \Psi^{-1}([\tilde{\mathbf{x}}_t; \tau])$    // Trajectory Alteration
9: **end for**
10: $\mathbf{x}_0 := \mathbf{x}_\theta(\tilde{\mathbf{x}}_t, t)$    // $\mathbf{x}_1 \to \mathbf{x}_0$
11: **return** $\mathbf{x}_0$

---

ize the boundary conditions $[\mathbf{x}_{t_0}; t_0]$ in $\tilde{p}_t(\tilde{\mathbf{x}}_t|\mathbf{x}_{t_0}, t_0, \mathbf{x}_0)$ of eq. (21) and $\Psi^{-1}([\mathbf{x}_{t_0}; t_0])$ of eq. (15) to any arbitrary condition pairs $[\tilde{\mathbf{x}}_t; \tau]$ and obtain the reverse process:

$$\tilde{p}([\tilde{\mathbf{x}}_{t-\Delta t}; \tau_\Delta]|[\tilde{\mathbf{x}}_t; \tau], \hat{\mathbf{x}}_0) =$$
$$\delta\left(\begin{bmatrix} \tilde{\mathbf{x}}_{t-\Delta t} \\ \tau_\Delta \end{bmatrix} - \begin{bmatrix} \mathbf{u}(\hat{\mathbf{x}}_0, \tau_\Delta)\hat{\mathbf{x}}_0 + \mathbf{v}(\hat{\mathbf{x}}_0, \tau_\Delta)\hat{\boldsymbol{\epsilon}} \\ \mathcal{T}(t - \Delta t, G(\hat{\mathbf{x}}_0, \hat{\boldsymbol{\epsilon}})) \end{bmatrix}\right), \tag{25}$$

where $\hat{\mathbf{x}}_0 = \mathbf{x}_\theta(\tilde{\mathbf{x}}_t, t)$ and $\tau_\Delta$ is the previous timestep of $\tau$ on the same rescaled trajectory.

Sampling from the reverse process is illustrated in algorithm 2. Similar to the sampling process of DDIM [Song et al., 2021a], it starts from the Gaussian noise, iteratively predicts the pseudo target $\hat{\mathbf{x}}_0$, and updates the reverse trajectory. However, since the $\tau$ and $\hat{\boldsymbol{\epsilon}}$ are mutually conditioned, we have to keep track of the $t, \tau, \tilde{\mathbf{x}}_t$, and $\hat{\boldsymbol{\epsilon}}$ during each iteration and split the update of $\hat{\boldsymbol{\epsilon}}$ into an asynchronous step (line 8). Because reverse trajectory keeps changing due to different pseudo targets $\hat{\mathbf{x}}_0$ predicted by learned neural networks, which brings severe instability, sometimes simply fixing the initial path (removing the line 8) exhibits better performance in experiments.

## 4   Language Modeling

Recent diffusion language models [Li et al., 2022, Gong et al., 2023b] inherit the embedding-rounding framework that a sentence with $n$ discrete tokens $W = [w_1, \ldots, w_n]$ is embedded to a continuous space via a trainable embedding layer $\text{EMB}(W) = [\text{EMB}(w_1), \ldots, \text{EMB}(w_n)]$. The vocabulary set is $K$ that $\forall w_n \in K$. Besides, the token embeddings are used as the target points $\mathbf{x}_0 = [\mathbf{x}_0^1, \ldots, \mathbf{x}_0^n]$, $\mathbf{x}_0^n = \text{EMB}(w_n)$, for continuous diffusion trajectories. Hence, generating tokens from embeddings is:

$$p(W|\mathbf{x}_0) = \sum_{i=1}^n p(w_i|\mathbf{x}_0^i) = \sum_{i=1}^n \frac{\exp(f(\mathbf{x}_0^i, w_i))}{\sum_{j \in K} \exp(f(\mathbf{x}_0^i, j))}, \tag{26}$$

where $f(\mathbf{x}, j) = \text{EMB}(j) \cdot \mathbf{x}$ is the dot production distance. It's also the function assessing the likelihood of point $\mathbf{x}$ inside the discrete area of $j$. The coefficient functions follow the DDPM [Ho et al., 2020], which are $\mathbf{u}(\mathbf{x}_0, t) = \sqrt{\bar{\alpha}_t}$ and $\mathbf{v}(\mathbf{x}_0, t) = \sqrt{1 - \bar{\alpha}_t}$. Besides, the objectives are

$$\mathcal{L}_\theta = \mathbb{E}_{W, t, \tilde{\mathbf{x}}_t}\left[\sum_{i=1}^n \|\text{EMB}(w_i) - \mathbf{x}_\theta(\tilde{\mathbf{x}}_t^i, t)\|^2/n\right] \tag{27}$$

Table 1: Result of BLEU scores on machine translation and ROUGE scores on text summarization.

| Models | IWSLT14 DE-EN BLEU (BLEU-1/2/3/4)⇑ | WMT14 EN-DE BLEU (BLEU-1/2/3/4)⇑ | WMT16 EN-RO BLEU (BLEU-1/2/3/4)⇑ | GIGAWORD ROUGE-1/2/L⇑ |
|---|---|---|---|---|
| *Auto-Regressive Modeling* | | | | |
| Transformers | 34.31 (67.3/41.6/27.9/19.1) | **28.01** (58.2/33.5/21.7/14.6) | 34.05 (63.1/39.9/27.6/19.6) | **37.57/18.90**/34.69 |
| Ours+Rerank | **35.02** (68.7/43.3/29.2/20.1) | 27.67 (57.9/33.2/21.4/14.3) | **34.33** (63.1/40.1/27.8/19.8) | 37.49/18.68/**34.82** |
| *Diffusion Process* | | | | |
| D3PM | 27.61 (65.4/37.7/22.8/14.2) | 22.94 (54.9/28.8/16.9/10.4) | 27.84 (59.8/34.9/22.1/14.5) | 33.92/14.96/31.72 |
| DiffuSeq | 28.78 ( - / - / - / - ) | 15.37 ( - / - / - / - ) | 25.45 ( - / - / - / - ) | 31.17/12.23/29.24 |
| SeqDiffuSeq | 30.03 ( - / - / - / - ) | 17.14 ( - / - / - / - ) | 26.17 ( - / - / - / - ) | 31.90/12.36/29.22 |
| Difformer | 31.58 (68.6/41.4/26.7/17.5) | 24.80 (58.7/32.0/19.7/12.5) | 30.08 (64.4/39.5/26.5/18.2) | 35.47/15.17/32.82 |
| SEDD | 31.87 (68.7/41.8/27.2/18.0) | 24.98 (59.2/32.4/20.1/12.9) | 29.38 (62.2/38.0/24.9/16.9) | 34.33/15.22/32.06 |
| Dinoiser | 31.91 (67.1/40.9/26.7/17.7) | 24.77 (57.2/31.0/19.0/12.0) | 31.49 (62.8/38.4/25.5/17.3) | 35.17/15.63/32.53 |
| Ours | **33.42** (68.0/42.0/27.7/18.6) | **26.69** (57.7/32.3/20.4/13.4) | **33.15** (63.4/39.9/27.4/19.2) | **36.44/16.09/33.56** |

and an additional rounding objective, which is commonly used in language modeling,

$$\mathcal{L}_r = -\log p_\theta(W|\mathbf{x}_0) = -\log p_\theta(W|\mathbf{x}_\theta(\tilde{\mathbf{x}}_t, t)). \tag{28}$$

The final training target is given by $\mathcal{L} = \mathcal{L}_\theta + \mathcal{L}_r$, where the $\mathbf{x}_0$ of the same token sequence $W$ keeps changing because the embedding layer EMB is trainable, which makes the model hard to be trained. Since previous work does not model discrete areas, a large number of noisy samples inside this area will make $\mathcal{L}_r$ too small to guide the training of the embedding layer, leading to a mode collapse.

**Experimental Setup** Datasets used for experiments include three translation tasks (IWSLT14 DE-EN [Cettolo et al., 2012], WMT14 EN-DE, and WMT16 EN-RO[1]) and one text summarization task (GIGAWORD [Rush et al., 2015]). We mainly follow the setting of Gao et al. [2022], which is inherited from previous non-auto-regressive text generation works [Gu et al., 2018, 2019, Ghazvininejad et al., 2019], where translation datasets are distilled [Kim and Rush, 2016]. Baselines are mainly continuous diffusion language models. DiffuSeq [Gong et al., 2023b] and SeqDiffuSeq [Yuan et al., 2022] are derived from Diffusion-LM [Li et al., 2022]. Difformer [Gao et al., 2022] and Dinoiser [Ye et al., 2023] are recent empirical studies highlighting that scaling up the noise is beneficial for language modeling. We also compare with discrete diffusion language models, including D3PM [Austin et al., 2021] and SEDD [Lou et al., 2023]. Since SEDD is a pre-trained language model, we configure its framework and train it from scratch specifically for our tasks. In addition, auto-regressive transformer [Vaswani et al., 2017] is still one of the most powerful architectures for language generation.

Our boundary conditional diffusion language model is constructed from Difformer [Gao et al., 2022], where the model configuration is *transformer-iwslt-de-en* in FAIRSEQ framework [Ott et al., 2019] for IWSLT14 DE-EN and *transformer-base* for other datasets. Sentences are tokenized with Byte-Pair Encoding [Sennrich et al., 2016] and evaluated by detokenized BLEU [Papineni et al., 2002] for machine translation and ROUGE [Lin, 2004] for summarization. During training, the diffusion step is $T = 2000$ and the confidence factor $r = 1$ for translation tasks since they have strong conditions, while $r = 0.5$ for summarization. Sentences are generated deterministically with 20 steps.

**Results** Performances are demonstrated in Table 1. Our approach achieves the state-of-the-art compared with continuous diffusion language models and outperforms the two discrete baselines on three machine translation and one text summarization tasks. Our method shows advantages, with a 73.6% significant improvement at most on WMT14 EN-DE, over DiffuSeq [Gong et al., 2023b] and SeqDiffuSeq [Yuan et al., 2022], which are two basic methods directly applying diffusion process to language modeling. Compared with recent strong diffusion language models like Difformer [Gao et al., 2022] and Dinoiser [Ye et al., 2023], which have deployed various effective noise scheduling strategies on diffusion processes from the empirical perspective, our model is still superior with at most 3.07 advancement of BLEU score on WMT16 EN-RO. This implies the effectiveness of modeling discrete priors. In addition, we illustrate the performance of auto-regressive modeling, where we use the transformer [Vaswani et al., 2017] to rerank the generated sentence candidates (7

---

[1] https://github.com/shawnkx/Fully-NAT

Table 3: Analysis on the training objectives.

| Objectives | $\mathbb{E}_{\tilde{\mathbf{x}}_t}\|\mathbf{x}_0 - \hat{\mathbf{x}}_0\|^2$ | $\mathbb{E}_{\tilde{\mathbf{x}}_t}\|\tilde{u}_t(\tilde{\mathbf{x}}_t|\mathbf{x}_0) - \tilde{u}_t(\tilde{\mathbf{x}}_t|\hat{\mathbf{x}}_0)\|^2$ | $\mathbb{E}_{\tilde{\mathbf{x}}_t}[p(\hat{\mathbf{x}}_0 \in C_{\mathbf{x}_0})]$ | BLEU |
|---|---|---|---|---|
| $\mathcal{L}_{\mathbf{x}_0}$ (eq. 24) | 8.44 | 1.56 | 51.81% | 33.42 |
| $\mathcal{L}_{\tilde{u}_t}$ | 8.41 | 1.55 | 52.34% | 33.49 |

length beam $\times$ 3 sentence beams) of our model. The reranked performance can even outperform transformers on IWSLT14 DE-EN and WMT16 EN-RO.

**Ablation** Our approach is a general framework applicable to almost all continuous diffusion models, providing them with discrete boundaries as priors. We choose Difformer [Gao et al., 2022] as the base model and follow the configurations. As proved in eq. (19), the original forward process will ignore the discrete priors although explicitly demonstrated. We conduct ablation experiments on the rescaling module. As illustrated in Table 2, our approach rescales the trajectory of both forward and reverse processes on Difformer. Only rescaling the forward trajectory is also effective but sub-optimal due to the inconsistent distribution during inference. Due to computational cost and fair comparison, our method leaves room for improvement. For example, replacing the forward trajectory with optimal transport in Flow Matching, $\mathbf{u}(\mathbf{x}_0, t) = 1 - t/T$ and $\mathbf{v}(\mathbf{x}_0, t) = t/T$, achieves better performance on WMT16.

Table 2: Ablation studies.

| Models | IWSLT14 | WMT16 |
|---|---|---|
| Base (Difformer) | 31.58 | 30.08 |
| + forward only | 33.02 | 32.86 |
| + forward & reverse | **33.42** | 33.15 |
| Optimal Transport | 32.77 | **33.65** |

**Analysis** Our training objective, eq. (24), is an upper bound of the eq. (2). We demonstrate the influence of this approximation in Table 3 on IWSLT14 DE-EN to reveal the thought of our formula. On the one hand, $\mathcal{L}_{\mathbf{x}_0}$ brings theoretical errors at a constant scale. On the other hand, $\mathcal{L}_{\mathbf{x}_0}$ mitigates some experimental errors from the neural networks. The first row $\mathcal{L}_{\mathbf{x}_0}$ is the objective we used in eq. (24) and the second row $\mathcal{L}_{\tilde{u}_t} = \mathbb{E}_{\{t,\mathbf{x}_0,\tilde{\mathbf{x}}_t\}}\left[\|\tilde{u}_t(\tilde{\mathbf{x}}_t|\mathbf{x}_\theta(\tilde{\mathbf{x}}_t, t)) - \frac{d\tilde{\mathbf{x}}_t}{dt}\|^2\right]$ is directly derived from the eq. (2). The first two columns represent the error expectations of $\mathbf{x}_0$ and $\tilde{u}_t$ on the test set. It is easy to observe that, with the dynamic coefficient $\frac{d\tau}{dt} = \frac{T - r \times G(\mathbf{x}_0, \boldsymbol{\epsilon})}{T}$ (appendix F), the value of $\mathbf{x}_0$'s error (8.44) is much larger than the $\tilde{u}_t$'s error (1.56). Therefore, $\mathcal{L}_{\mathbf{x}_0}$ is beneficial for reducing the impact of the prediction error from the neural network. The third column in Table 3 illustrates the one-step accuracy of predicting $\mathbf{x}_0$ and the fourth column is the BLEU score on the test set. Experimental results show that optimizing the upper bound has a negligible impact on the final performance (only a $0.2\%$ drop of the BLEU score), while can improve the efficiency of the loss calculation during the training phase.

## 5 Discrete Image Generation

Image pixels are usually treated as real numbers in continuous space since adjacent pixel values exhibit linear continuity.They are essentially discrete and quantized data with a finite state space, such as 256 states in RGB format. We utilize two discrete image representations. One is binary coding provided by Bit Diffusion [Chen et al., 2023b] that converts a sub-pixel with 256 integers to a 8-bit binary code. It is more efficient as it stores ordinal relationships, but the representation space it constructs will be sparse. Another is pixel embedding, which is a more discrete form of representation because the relationships between pixels are thoroughly broken down and reconstructed by learning the embedding representation. Each pixel is regarded as a one-hot vector and transformed with an embedding layer EMB as used in language. Furthermore, we design an intermediate state to demonstrate the correlation between discreteness and modeling difficulty, which is initializing a fixed embedding with binary coding. The optimization target for binary coding is the MSE loss, and pixel embeddings take the same objective as in language.

**Experimental Setup** We use CIFAR-10 [Krizhevsky et al., 2009] for discrete image generation. The evaluation metric is FID [Heusel et al., 2017], which compares 50K generated samples with the training set. Our image generation model is constructed on Bit Diffusion [Chen et al., 2023b], where the architecture is U-Net [Ronneberger et al., 2015] with 3 stages, 256 channels and 3 residual blocks

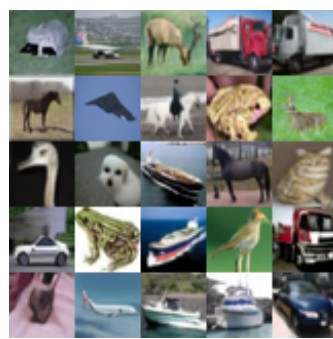 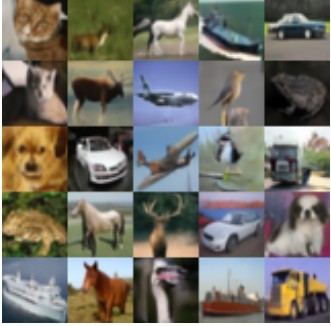 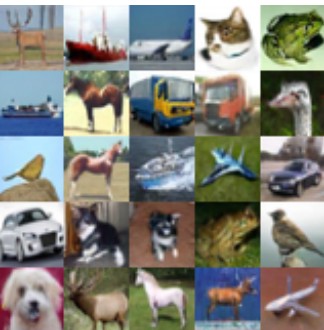

| (A) Bit Diffusion *repro* (FID 10.37) | (B) DDIM (FID 4.04) | (C) Ours (FID 3.86) |

Figure 3: Generated images of Bit Diffusion *repro*, DDIM, and Ours on CIFAR-10.

per stage. Diffusion steps are $T = 1000$ for both the training and inference stages. The model is trained for $1.5M$ steps with the learning rate of $1e\text{-}4$ and batch size of $128$. Since the training script and detailed hyperparameters of Bit Diffusion are not available, we have to reproduce it by ourselves and our boundary conditional diffusion model shares exactly the same configuration. Our confidence factors are $r = 0.5$ for all three settings. Other baselines include D3PM [Austin et al., 2021] and $\tau$LDR [Campbell et al., 2022] which are discrete diffusion models. SDDM [Sun et al., 2023] utilizes vector quantization from VQ-GAN [Esser et al., 2021] as a continuous space for discrete data. We also compare with DDPM [Ho et al., 2020] and DDIM [Song et al., 2021a] on continuous pixels.

**Results** For binary coding, as shown in Table 4, our approach outperforms the reproduced Bit Diffusion and attains competitive results to state-of-the-art models. For pixel embedding where ordinal information is deconstructed and reconstituted, our method exhibits a notable improvement of $3.81$ FID score over replicated Bit Diffusion. Moreover, in the case of categorical pixels, this advantage increases to $8.25$, positioning our approach with trainable embedding as a new state-of-the-art solution. Additionally, as deterministic diffusion processes, our model with binary coding can slightly exceed the performance of DDIM, where the generated samples are in Figure 3.

**Analysis** We analyze the influence of the confidence factor $r$ in Table 5. The factor $r$ is selected from $[0, 0.2, 0.3, 0.5]$, where $r = 0$ is the reproduced Bit Diffusion that discards the discrete priors. As the confidence factor increases, the impact of discreteness gradually improves, simultaneously enhancing the model's performance across all three settings. Since there is no guidance for unconditional image generation, we do not use a larger factor to prevent mode collapses.

Table 4: FID scores on CIFAR-10.

| Models | CIFAR-10 (FID ⇓) | | |
|---|---|---|---|
| | 200K | 500K | Final |
| *Continuous Pixels* | | | |
| **DDPM** | - | - | 3.17 |
| **DDIM** | - | - | 4.04 |
| *Discrete Ordinal Pixels* | | | |
| **D3PM** GAUSS | - | - | 7.34 |
| $\tau$**LDR**-0 | - | - | 8.10 |
| $\tau$**LDR**-10 | - | - | 3.74 |
| BINARY CODING (UINT8): | | | |
| **Bit Diffusion** | - | - | 3.48 |
| **Bit Diffusion** *repro* | 22.12 | 13.23 | 10.37 |
| **Ours** | 8.17 | 5.03 | **3.86** |
| FIXED EMBEDDING: | | | |
| **Bit Diffusion** *repro* | 19.69 | 16.61 | 12.96 |
| **Ours** | 12.32 | 10.09 | **9.15** |
| *Categorical Pixels* | | | |
| **D3PM** UNIFORM | - | - | 51.27 |
| **D3PM** ABSORBING | - | - | 30.97 |
| VECTOR QUANTIZATION: | | | |
| **D3PM**-VQ | - | - | 16.47 |
| $\tau$**LDR**-VQ | - | - | 40.06 |
| **SDDM**-VQ | - | - | 12.23 |
| TRAINABLE EMBEDDING: | | | |
| **Bit Diffusion** *repro* | 33.09 | 27.21 | 19.26 |
| **Ours** | 21.17 | 15.32 | **10.99** |

## 6 Related Work

**Discrete Modeling** Auto-regressive models have demonstrated a domination over discrete modeling, especially for text generation [Vaswani et al., 2017, Brown et al., 2020, Achiam et al., 2023]. However, the computation

Table 5: Confidence factors.

| Models | r = 0 | 0.2 | 0.3 | 0.5 |
|---|---|---|---|---|
| BINARY CODING | 10.37 | 7.39 | 5.33 | 3.86 |
| FIXED EMBEDDING | 12.96 | 11.35 | 10.80 | 9.15 |
| TRAINABLE EMBEDDING | 19.26 | 15.32 | 11.56 | 10.99 |

cost increases drastically as the size of sentence length or the image resolution increases. Diffusion models [Sohl-Dickstein et al., 2015, Ho et al., 2020, Dhariwal and Nichol, 2021, Saharia et al., 2022] can generate data in parallel, but are tailored for continuous problems. To generalize diffusion models for discrete data, the most straightforward methods define discrete processes in discrete spaces [Sohl-Dickstein et al., 2015, Hoogeboom et al., 2021b, Austin et al., 2021, Campbell et al., 2022, Zhang et al., 2023, Sun et al., 2023, Lou et al., 2023], which will be bothered by large number of discrete status. Besides, a simplified version of discrete diffusion processes is recently used in language modeling [He et al., 2023, Chen et al., 2023a]. Approaches in another line argue to located discrete data in continuous spaces, which is more flexible and efficient, with the mapping functions including binary bits [Chen et al., 2023b] and embeddings [Li et al., 2022, Gong et al., 2023b,a, Yuan et al., 2022, Gulrajani and Hashimoto, 2023, Han et al., 2023]. Other generative models adapted for discrete modeling includes Variational Autoencoders [Kingma and Welling, 2014], Generative Adversarial Networks [Hjelm et al., 2018, Fedus et al., 2018], and Normalizing Flows [Lindt and Hoogeboom, 2021, Hoogeboom et al., 2021a, Tan et al., 2022].

**Diffusion Models with Deterministic Trajectory**    Deterministic diffusion process is usually used in the inference stage to speed up sampling, where DDIM [Song et al., 2021a] derives a serial of non-Markovian diffusion processes and the deterministic one is a special case from this implicit perspective. Additionally, deterministic diffusion processes can be converted to ordinary differential equations [Song et al., 2021b], which is utilized by recent sampling acceleration approaches such as DEIS [Zhang and Chen, 2023] and DPM-Solvers [Lu et al., 2022b,a, Zheng et al., 2023]. Our approach requires a deterministic forward trajectory to eliminate the randomness between the boundary point and sampled point. Flow matching [Liu, 2022, Lipman et al., 2023, Albergo and Vanden-Eijnden, 2023, Liu et al., 2023] is a collection of generative models that employ ordinary differential equations to facilitate both forward and reverse processes. They can be regarded as generally equivalent to Diffusion models. Therefore, we extend the framework of flow matching for our method.

# 7    Conclusion

We studied the gap between discrete modeling and continuous spaces, focusing on the inconsistency between probability density contours learned by continuous diffusion models and discrete boundaries. We have proposed a novel and general approach to address this issue by enabling continuous diffusion models to be conditioned on discrete priors, which is achieved via discrete boundary estimation and trajectory rescaling. An important limitation is that our method is designed for continuous diffusion models, where discrete diffusion models constructed specially on the discrete state space would not encounter the problem. However, discrete diffusion models also possess their own shortcomings, and the practical applications of continuous diffusion models are more extensive. We believe that our method has the potential to advance the development of unified and general diffusion models. By bridging the gap between discrete and continuous modeling, we hope to inspire new possibilities for modeling complex systems and phenomena.

# Acknowledgements

Bing Qin is the corresponding author of this work, We thank the anonymous reviewers for their insightful comments. This work was supported by the National Natural Science Foundation of China (NSFC) (U22B2059, grant 62276078), the Key R&D Program of Heilongjiang via grant 2022ZX01A32, the International Cooperation Project of PCL, PCL2022D01 and the Fundamental Research Funds for the Central Universities (Grant No.HIT.OCEF.2023018).

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

## A  Stopping Time for Forward Process

The forward diffusion process $\mathbf{X} = \{\mathbf{x}_n, n \geq 0\}$ is a markovian stochastic process with a transition probability $p(\mathbf{x}_i|\mathbf{x}_{i-1}) = \mathcal{N}\left(\mathbf{x}_i; \sqrt{\alpha_i}\mathbf{x}_{i-1}, (1 - \alpha_i)\mathbf{I}\right)$. And a stopping time $t_0$ with respect to $\mathbf{X}$ is a random time such that for each $n \geq 0$, the event $\{t_0 = n\}$ is completely determined by the total information known up to time $n$, $\{\mathbf{x}_0, \ldots, \mathbf{x}_n\}$. Suppose the random variables $\{\mathbf{x}_n\}$ are in a one-dimensional space and the forward process starts with $\mathbf{x}_0 = 0$. Besides, let $A, \mathbf{x}_0 \in A$ be the discrete area belonging to $\mathbf{x}_0$ that for each points in area $A$ will be regarded as $\mathbf{x}_0$ during data generation. Our expected stopping time is defined as:

$$t_0 = \min\{n \geq 0, \mathbf{x}_n \notin A\},$$

which represents the first time $\mathbf{x}_n$ leaves area $A$. We can write the probability of stopping time as:

$$
\begin{aligned}
P(t_0 = 0) &= P(\mathbf{x}_0 \notin A) = 0 \\
P(t_0 = 1) &= P(\mathbf{x}_0 \in A, \mathbf{x}_1 \notin A) \\
&= \int_{\mathbf{x}_1 \notin A} \mathcal{N}\left(\mathbf{x}_1; \sqrt{\alpha_1}\mathbf{x}_0, (1 - \alpha_1)\mathbf{I}\right) d\mathbf{x}_1 \\
P(t_0 = 2) &= P(\mathbf{x}_0 \in A, \mathbf{x}_1 \in A, \mathbf{x}_2 \notin A) \\
&= P(\mathbf{x}_0 \in A, \mathbf{x}_1 \in A) \times P(\mathbf{x}_2 \notin A|\mathbf{x}_1 \in A) \\
&= \int_{\mathbf{x}_2 \notin A} \left[ \int_{\mathbf{x}_1 \in A} \mathcal{N}\left(\mathbf{x}_1; \sqrt{\alpha_1}\mathbf{x}_0, (1 - \alpha_1)\mathbf{I}\right) \times \right. \\
&\quad \left. \mathcal{N}\left(\mathbf{x}_2; \sqrt{\alpha_2}\mathbf{x}_1, (1 - \alpha_2)\mathbf{I}\right) d\mathbf{x}_1 \right] d\mathbf{x}_2 \\
&\cdots\cdots \\
P(t_0 = n) &= P(\mathbf{x}_0 \in A, \ldots, \mathbf{x}_{n-1} \in A, \mathbf{x}_n \notin A) \\
&= \int_{\mathbf{x}_n \notin A} \int_{\mathbf{x}_{\leq n} \in A} \prod_{i=1}^{n-1} \mathcal{N}\left(\mathbf{x}_i; \sqrt{\alpha_i}\mathbf{x}_{i-1}, (1 - \alpha_i)\mathbf{I}\right) d\mathbf{x}_{1:n}.
\end{aligned}
$$

Since the diffusion process is established in continuous space, calculating the probability of the stopping time requires integrating over each intermediate state $\mathbf{x}_{1:n-1}$, rather than a simple state transfer as in the discrete space. Hence, directly obtain the stopping time is intractable. Additionally, even if we are able to get probability of the stopping time, we can only get a distribution over the time dimension, without knowing the exact time of $\mathbf{x}_n$ leaving area $A$. Therefore, we need to eliminate randomness from the state transition $\mathbf{x}_{i-1} \rightarrow \mathbf{x}_i$ and find a deterministic forward trajectory to estimate the stopping time.

## B  Properties of Dirac Delta Function

There are several useful properties of Dirac delta function:

- **Symmetry Property:** $\delta(-x) = \delta(x)$

- **Scaling Property:** $\delta(ax) = \frac{\delta(x)}{|a|}$

- **Translation Property:** $\int f(x)\delta(x - a)dx = f(a)$

## C  Bridging Flow Matching and DDPM

In this work, we utilizes the framework of Flow Matching to model the diffusion processes, where the forward process is defined by flow functions in eq. (5). Although having different mathematical forms, it is essentially equivalent to traditional diffusion processes. Here, we provide an alternative form from the perspective of state transfer $p_t(\mathbf{x}_t|\mathbf{x}_{t-1})$.

## C.1 Deterministic Forward Process

Equation (5) gives the definition $p_t(\mathbf{x}_t|\mathbf{x}_0) = [\psi_t]_*\pi(\mathbf{x})$, where $\psi_t(\mathbf{x}) = \mathbf{u}_t\mathbf{x}_0 + \mathbf{v}_t\mathbf{x}$. Here we provide the equivalent derivation of $p_t(\mathbf{x}_t|\mathbf{x}_0)$ from the perspective of diffusion processes:

$$
\begin{aligned}
p_t(\mathbf{x}_t|\mathbf{x}_0) &= \int p_t(\mathbf{x}_{1:t}|\mathbf{x}_0)\mathrm{d}\mathbf{x}_{1:t-1} \\
&= \int p(\mathbf{x}_1|\mathbf{x}_0)\prod_{s=2}^{t}p_s(\mathbf{x}_s|\mathbf{x}_{s-1},\mathbf{x}_0)\mathrm{d}\mathbf{x}_{1:t-1},
\end{aligned}
\tag{29}
$$

where $p(\mathbf{x}_1|\mathbf{x}_0) = \mathcal{N}(\mathbf{u}_1\mathbf{x}_0, \mathbf{v}_1^2\mathbf{I})$ is the first step of the forward process at which the global noise is introduced into the forward trajectory. The state transfer probability of forward process $p_s(\mathbf{x}_s|\mathbf{x}_{s-1},\mathbf{x}_0) = \delta(\mathbf{x}_s - \mathbf{u}_s\mathbf{x}_0 - \mathbf{v}_s\psi_{s-1}^{-1}(\mathbf{x}_{s-1}))$ is a Dirac delta function. Therefore,

$$
\begin{aligned}
p_t(\mathbf{x}_t|\mathbf{x}_0) &= \int \prod_{s=3}^{t}p_s(\mathbf{x}_s|\mathbf{x}_{s-1},\mathbf{x}_0)\mathbf{dx}_{2:t-1} \\
&\times \underbrace{\int p_2(\mathbf{x}_2|\mathbf{x}_1,\mathbf{x}_0)p(\mathbf{x}_1|\mathbf{x}_0)\mathrm{d}\mathbf{x}_1}_{Q_1},
\end{aligned}
\tag{30}
$$

where we denote the integral of $\mathbf{x}_1$ as $Q_1$. Based on

$$
\begin{aligned}
Q_0 &= q(\mathbf{x}_1|\mathbf{x}_0) = \mathcal{N}(\mathbf{u}_1\mathbf{x}_0, \mathbf{v}_1^2\mathbf{I}) \\
q_2(\mathbf{x}_2|\mathbf{x}_1,\mathbf{x}_0) &= \delta\left(\mathbf{x}_2 - \mathbf{u}_2\mathbf{x}_0 - \mathbf{v}_2\psi_1^{-1}(\mathbf{x}_1)\right) \\
&= \delta\left[\mathbf{x}_2 - \frac{\mathbf{v}_2}{\mathbf{v}_1}\mathbf{x}_1 - \left(\mathbf{u}_2 - \frac{\mathbf{v}_2\mathbf{u}_1}{\mathbf{v}_1}\right)\mathbf{x}_0\right] \\
&= \delta\left[\mathbf{x}_1 - \frac{\mathbf{v}_1}{\mathbf{v}_2}\mathbf{x}_2 - \left(\mathbf{u}_1 - \frac{\mathbf{v}_1\mathbf{u}_2}{\mathbf{v}_2}\right)\mathbf{x}_0\right]
\end{aligned}
\tag{31}
$$

(Symmetry Property of Dirac Delta Function)

and the **Translation Property** of the Dirac delta function, we can calculate $Q_1$ as:

$$
\begin{aligned}
Q_1 &= \int \underbrace{p_2(\mathbf{x}_2|\mathbf{x}_1,\mathbf{x}_0)}_{\delta(x-a)}\underbrace{p(\mathbf{x}_1|\mathbf{x}_0)}_{f(x)}\mathrm{d}\mathbf{x}_1, \\
&\text{where } \begin{cases} x : \mathbf{x}_1 \\ a : \frac{\mathbf{v}_1}{\mathbf{v}_2}\mathbf{x}_2 + \left(\mathbf{u}_1 - \frac{\mathbf{v}_1\mathbf{u}_2}{\mathbf{v}_2}\right)\mathbf{x}_0 \end{cases} \\
&\implies Q_1 = \mathcal{N}(\mathbf{u}_2\mathbf{x}_0, \mathbf{v}_2^2\mathbf{I}.)
\end{aligned}
\tag{32}
$$

Then we can continue the deviation of $p_t(\mathbf{x}_t|\mathbf{x}_0)$ as:

$$
\begin{aligned}
p_t(\mathbf{x}_t|\mathbf{x}_0) &= \int Q_0\prod_{s=2}^{t}p_s(\mathbf{x}_s|\mathbf{x}_{s-1},\mathbf{x}_0)\mathrm{d}\mathbf{x}_{1:t-1} \\
&= \int Q_1\prod_{s=3}^{t}p_s(\mathbf{x}_s|\mathbf{x}_{s-1},\mathbf{x}_0)\mathrm{d}\mathbf{x}_{2:t-1} \\
&= \cdots\cdots \\
&= \int p_t(\mathbf{x}_t|\mathbf{x}_{t-1})Q_{t-2}\mathrm{d}\mathbf{x}_{t-1} \\
&= Q_{t-1} = \mathcal{N}(\mathbf{u}_t\mathbf{x}_0, \mathbf{v}_t^2\mathbf{I})
\end{aligned}
\tag{33}
$$

Therefore, the probability distribution of $\mathbf{x}_t$ conditioned on $\mathbf{x}_0$ follows a Gaussian distribution $\mathcal{N}(\mathbf{u}_t\mathbf{x}_0, \mathbf{v}_t^2\mathbf{I})$, which is the same as in original DDPMs when the coefficient functions are defined as $\mathbf{u}_t = \sqrt{\bar{\alpha}_t}$ and $\mathbf{v}_t = \sqrt{1 - \bar{\alpha}_t}$. This provides an important benefit that the Flow Matching and diffusion models share the same training procedure.

## C.2  Deterministic Reverse Process

The reverse tranfer probability follows Bayes' rule:

$$
\begin{aligned}
p(\mathbf{x}_{t-1}|\mathbf{x}_t, \mathbf{x}_0) &= p_t(\mathbf{x}_t|\mathbf{x}_{t-1}, \mathbf{x}_0)\frac{p_{t-1}(\mathbf{x}_{t-1}|\mathbf{x}_0)}{p_t(\mathbf{x}_t|\mathbf{x}_0)} \\
&= \frac{p_{t-1}(\mathbf{x}_{t-1}|\mathbf{x}_0)}{p_t(\mathbf{x}_t|\mathbf{x}_0)} \times \delta\left[\mathbf{x}_t - \frac{\mathbf{v}_t}{\mathbf{v}_{t-1}}\mathbf{x}_{t-1} - \left(\mathbf{u}_t - \frac{\mathbf{v}_t\mathbf{u}_{t-1}}{\mathbf{v}_{t-1}}\right)\mathbf{x}_0\right].
\end{aligned}
\tag{34}
$$

Since Dirac delta function has another form of

$$
\delta(x) = \begin{cases} +\infty, x = 0 \\ \quad 0, x \neq 0 \end{cases},
\tag{35}
$$

and $p_t(\mathbf{x}_t|\mathbf{x}_0) > 0$, $p_{t-1}(\mathbf{x}_{t-1}|\mathbf{x}_t) > 0$, we have

$$
\begin{aligned}
p(\mathbf{x}_{t-1}|\mathbf{x}_t, \mathbf{x}_0) &= p_t(\mathbf{x}_t|\mathbf{x}_{t-1}, \mathbf{x}_0)\frac{p_{t-1}(\mathbf{x}_{t-1}|\mathbf{x}_0)}{p_t(\mathbf{x}_t|\mathbf{x}_0)} \\
&= \begin{cases} +\infty \times \overbrace{\dfrac{p_{t-1}(\mathbf{x}_{t-1}|\mathbf{x}_0)}{p_t(\mathbf{x}_t|\mathbf{x}_0)}}^{>0}, \quad \mathbf{x}_t = \left[\dfrac{\mathbf{v}_t}{\mathbf{v}_{t-1}}\mathbf{x}_{t-1} + \left(\mathbf{u}_t - \dfrac{\mathbf{v}_t\mathbf{u}_{t-1}}{\mathbf{v}_{t-1}}\right)\mathbf{x}_0\right] \\[2em] \qquad\quad 0, \quad \mathbf{x}_t \neq \left[\dfrac{\mathbf{v}_t}{\mathbf{v}_{t-1}}\mathbf{x}_{t-1} + \left(\mathbf{u}_t - \dfrac{\mathbf{v}_t\mathbf{u}_{t-1}}{\mathbf{v}_{t-1}}\right)\mathbf{x}_0\right] \end{cases} \\[2em]
&\simeq \begin{cases} +\infty, \ \mathbf{x}_{t-1} = \left[\dfrac{\mathbf{v}_{t-1}}{\mathbf{v}_t}\mathbf{x}_t + \left(\mathbf{u}_{t-1} - \dfrac{\mathbf{u}_t\mathbf{v}_{t-1}}{\mathbf{v}_t}\right)\mathbf{x}_0\right] \\[2em] \quad 0, \ \mathbf{x}_{t-1} \neq \left[\dfrac{\mathbf{v}_{t-1}}{\mathbf{v}_t}\mathbf{x}_t + \left(\mathbf{u}_{t-1} - \dfrac{\mathbf{u}_t\mathbf{v}_{t-1}}{\mathbf{v}_t}\right)\mathbf{x}_0\right] \end{cases} \\[2em]
&= \delta\left[\mathbf{x}_{t-1} - \frac{\mathbf{v}_{t-1}}{\mathbf{v}_t}\mathbf{x}_t - \left(\mathbf{u}_{t-1} - \frac{\mathbf{u}_t\mathbf{v}_{t-1}}{\mathbf{v}_t}\right)\mathbf{x}_0\right] \\[1em]
&= \lim_{\sigma \to 0}\mathcal{N}\left(\frac{\mathbf{v}_{t-1}}{\mathbf{v}_t}\mathbf{x}_t + \left(\mathbf{u}_{t-1} - \frac{\mathbf{u}_t\mathbf{v}_{t-1}}{\mathbf{v}_t}\right)\mathbf{x}_0, \sigma^2\mathbf{I}\right).
\end{aligned}
\tag{36}
$$

## C.3  Deterministic Optimization Objective

We first include the derivation of the variational bound for diffusion models provided by Sohl-Dickstein et al. [2015]. The probability the generative model assigns to the data is:

$$
\begin{aligned}
p(\mathbf{x}_0) &= \int p(\mathbf{x}_{0:T})d\mathbf{x}_{1:T} \\
&= \int p(\mathbf{x}_{0:T})\frac{p_T(\mathbf{x}_{1:T}|\mathbf{x}_0)}{p_T(\mathbf{x}_{1:T}|\mathbf{x}_0)}d\mathbf{x}_{1:T} \\
&= \int p_T(\mathbf{x}_{1:T}|\mathbf{x}_0)\frac{p(\mathbf{x}_{0:T})}{p_T(\mathbf{x}_{1:T}|\mathbf{x}_0)}d\mathbf{x}_{1:T} \\
&= \int p_T(\mathbf{x}_{1:T}|\mathbf{x}_0)p(\mathbf{x}_T)\prod_{t=1}^{T}\frac{p(\mathbf{x}_{t-1}|\mathbf{x}_t)}{p_t(\mathbf{x}_t|\mathbf{x}_{t-1})}d\mathbf{x}_{1:T}.
\end{aligned}
\tag{37}
$$

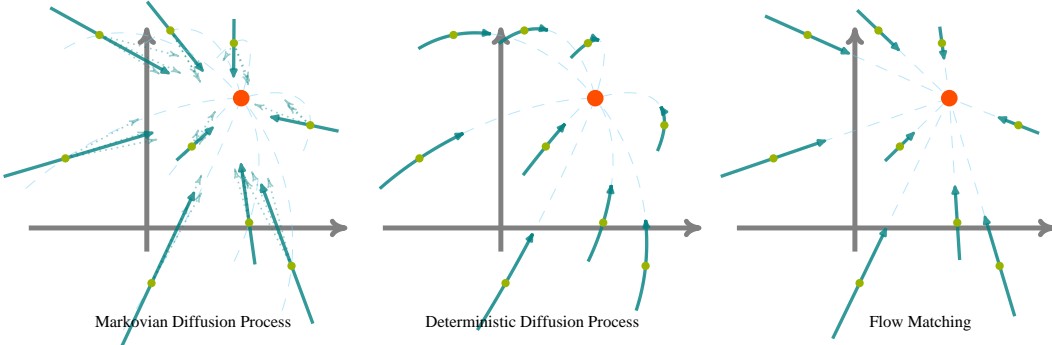

Figure 4: We demonstrate the trajectory differences among Markovian Diffusion Process, Deterministic Diffusion and Flow Matching.

Training amounts to minimizing the negative log likelihood:

$$
\begin{aligned}
\mathcal{L} &= -\int p(\mathbf{x}_0) \log p(\mathbf{x}_0) \mathrm{d}\mathbf{x}_0 \\
&= -\int p(\mathbf{x}_0) \log \left[ \int p_T(\mathbf{x}_{1:T}|\mathbf{x}_0) p(\mathbf{x}_T) \prod_{t=1}^{T} \frac{p(\mathbf{x}_{t-1}|\mathbf{x}_t)}{p_t(\mathbf{x}_t|\mathbf{x}_{t-1})} \mathrm{d}\mathbf{x}_{1:T} \right] \mathrm{d}\mathbf{x}_0 \\
&\leq -\int p_T(\mathbf{x}_{0:T}) \log \left[ p(\mathbf{x}_T) \prod_{t=1}^{T} \frac{p(\mathbf{x}_{t-1}|\mathbf{x}_t)}{p_t(\mathbf{x}_t|\mathbf{x}_{t-1})} \right] \mathrm{d}\mathbf{x}_{0:T} \\
&= \mathbb{E}_{p_T(\mathbf{x}_{0:T})} \left[ -\log p(\mathbf{x}_T) + \sum_{t=1}^{T} \log \frac{p_t(\mathbf{x}_t|\mathbf{x}_{t-1})}{p(\mathbf{x}_{t-1}|\mathbf{x}_t)} \right] \\
&= \mathbb{E}_{p_T} \left[ \log \frac{p_T(\mathbf{x}_T|\mathbf{x}_0)}{p(\mathbf{x}_T)} - \log p(\mathbf{x}_0|\mathbf{x}_1) + \sum_{t=2}^{T} \log \frac{p(\mathbf{x}_{t-1}|\mathbf{x}_t, \mathbf{x}_0)}{p(\mathbf{x}_{t-1}|\mathbf{x}_t)} \right] \\
&= \mathbb{E}_{p_T} \left[ \underbrace{D_{\mathrm{KL}}(p_T(\mathbf{x}_T|\mathbf{x}_0)||p(\mathbf{x}_T))}_{\mathcal{L}_T} \underbrace{- \log p(\mathbf{x}_0|\mathbf{x}_1)}_{\mathcal{L}_0} + \sum_{t=2}^{T} \underbrace{D_{\mathrm{KL}}(p(\mathbf{x}_{t-1}|\mathbf{x}_t, \mathbf{x}_0)||p(\mathbf{x}_{t-1}|\mathbf{x}_t))}_{\mathcal{L}_{t-1}} \right]
\end{aligned}
$$

where $\mathcal{L}_T$ is usually ignored as a constant and $p(\mathbf{x}_{t-1}|\mathbf{x}_t)$ is parameterized with a neural network $p_\theta(\mathbf{x}_{t-1}|\mathbf{x}_t)$ to approximate the conditioned probability distributions in the reverse process. Since $p(\mathbf{x}_{t-1}|\mathbf{x}_t, \mathbf{x}_0) = \lim_{\sigma \to 0} \mathcal{N}\left( \frac{\mathbf{v}_{t-1}}{\mathbf{v}_t}\mathbf{x}_t + \left( \mathbf{u}_{t-1} - \frac{\mathbf{u}_t \mathbf{v}_{t-1}}{\mathbf{v}_t}\mathbf{x}_0 \right), \sigma^2 \mathbf{I} \right)$, the parameterized $p_\theta(\mathbf{x}_{t-1}|\mathbf{x}_t)$ can take the same form $\mathcal{N}(\boldsymbol{\mu}_\theta(\mathbf{x}_t, t), \sigma_t^2 \mathbf{I})$ because the Dirac delta function is a special case of Gaussian distribution and the KL divergence of two Gaussians can be simplified. Finally, the training objective for the deterministic diffusion process is divided as:

$$
\mathcal{L} = \begin{cases} \mathcal{L}_T : \text{a constant} \\ \mathcal{L}_0 : -\log \delta\left(\mathbf{x}_0 - \mathbf{x}_\theta(\mathbf{x}_1, 1)\right) \\ \mathcal{L}_{t-1} : c\|\mathbf{x}_0 - \mathbf{x}_\theta(\mathbf{x}_t, t)\|^2 + \lim_{\sigma \to 0} \log \frac{\sigma_t}{\sigma} \\ c = \frac{1}{2\sigma_t^2} \left( \mathbf{u}_{t-1} - \frac{\mathbf{u}_t \mathbf{v}_{t-1}}{\mathbf{v}_{t-1}} \right)^2, \end{cases} \tag{38}
$$

where the simplified version $\|\mathbf{x}_0 - \mathbf{x}_\theta(\mathbf{x}_t, t)\|^2$ is the same as DDPMs but with different coefficients.

## D  Different Diffusion Trajectories

We illustrate the trajectories of different diffusion processes in Figure 4. The forward and reverse generation for the Markovian diffusion process is:

$$\begin{cases} \mathbf{x}_t = \sqrt{\bar{\alpha}_t}\mathbf{x}_0 + \sqrt{1 - \bar{\alpha}_t}\boldsymbol{\epsilon}_t \\ \mathbf{x}_{t-1} = \dfrac{\sqrt{\bar{\alpha}_{t-1}}(1 - \alpha_t)}{1 - \bar{\alpha}_t}\mathbf{x}_0 + \dfrac{\sqrt{\alpha_t}(1 - \bar{\alpha}_{t-1})}{1 - \bar{\alpha}_t}\mathbf{x}_t \\ \qquad\quad + \dfrac{\sqrt{(1 - \bar{\alpha}_{t-1})(1 - \alpha_t)}}{\sqrt{1 - \bar{\alpha}_t}}\boldsymbol{\epsilon}_{t-1}. \end{cases} \tag{39}$$

The deterministic diffusion process:

$$\begin{cases} \mathbf{x}_t = \sqrt{\bar{\alpha}_t}\mathbf{x}_0 + \sqrt{1 - \bar{\alpha}_t}\boldsymbol{\epsilon} \\ \mathbf{x}_{t-1} = \left(\sqrt{\bar{\alpha}_{t-1}} - \dfrac{\sqrt{\bar{\alpha}_t(1 - \bar{\alpha}_{t-1})}}{\sqrt{1 - \bar{\alpha}_t}}\right)\mathbf{x}_0 \\ \qquad\quad + \dfrac{\sqrt{1 - \bar{\alpha}_{t-1}}}{\sqrt{1 - \bar{\alpha}_t}}\mathbf{x}_t. \end{cases} \tag{40}$$

The deterministic flow matching with optimal transport:

$$\begin{cases} \mathbf{x}_t = (1 - \dfrac{t}{T})\mathbf{x}_0 + \dfrac{t}{T}\boldsymbol{\epsilon} \\ \mathbf{x}_{t-1} = \dfrac{1}{t}\mathbf{x}_0 + \dfrac{t - 1}{t}\mathbf{x}_t. \end{cases} \tag{41}$$

## E  Details of the Function $G$

Equation (13) defines the function $G(\mathbf{x}, \boldsymbol{\epsilon})$ as the inversion of coefficient function.

**Flow Matching**   The coefficient is $\mathbf{u}_t = 1 - t/T$, where $t = T \times (1 - \mathbf{u}_t)$. Therefore,

$$G(\mathbf{x}_0, \boldsymbol{\epsilon}) = t_0 = T \times (1 - \mathbf{u}_{t_0}) = T \left/ \left(1 + \frac{f(\boldsymbol{\epsilon}, \mathcal{J}) - f(\boldsymbol{\epsilon}, \mathcal{I})}{f(\mathbf{x}_0, \mathcal{I}) - f(\mathbf{x}_0, \mathcal{J})}\right)\right. \tag{42}$$

**Diffusion**   The coefficient for Variance Exploding is $\mathbf{v}_T = \sigma_0 \left(\frac{\sigma_T}{\sigma_0}\right)^{\frac{t}{T}}$, where $t = T \times \frac{\log \mathbf{v}_t - \log \sigma_0}{\log \sigma_T - \log \sigma_0}$.

$$G(\mathbf{x}_0, \boldsymbol{\epsilon}) = t_0 = \mathbf{v}_{t_0} = T \times \frac{\log \mathbf{v}_t - \log \sigma_0}{\log \sigma_T - \log \sigma_0} = T \times \frac{\log \frac{f(\boldsymbol{\epsilon}, \mathcal{J}) - f(\boldsymbol{\epsilon}, \mathcal{I})}{f(\mathbf{x}_0, \mathcal{I}) - f(\mathbf{x}_0, \mathcal{J})} - \log \sigma_0}{\log \sigma_T - \log \sigma_0}. \tag{43}$$

For Variance Preserving, the function $G(\mathbf{x}_0, \boldsymbol{\epsilon})$ is more difficult to calculate since $\mathbf{u}_t = \sqrt{\bar{\alpha}_t}$, where $\bar{\alpha} = \prod_{i=1}^{t} \alpha_i$, $\alpha_t = 1 - \beta_t$, and $\beta_t$ is also influenced by noise schedulers. This makes $G(\mathbf{x}_0, \boldsymbol{\epsilon})$ hard to calculate. Fortunately, we can bypass this function and provide the corresponding pseudo code.

## F  Details of the Training Objective

The rescaled vector field is calculated as:

$$\begin{aligned} \tilde{u}_t &= \frac{\mathrm{d}\tilde{\mathbf{x}}_t}{\mathrm{d}t} = \frac{\mathrm{d}\tilde{\mathbf{x}}_t}{\mathrm{d}\tau}\frac{\mathrm{d}\tau}{\mathrm{d}t} \\ &= \left[\mathbf{u}'\left(\mathbf{x}_0, \tau\right)\mathbf{x}_0 + \mathbf{v}'(\mathbf{x}_0, \tau)\boldsymbol{\epsilon}\right]\frac{T - r \times G(\mathbf{x}_0, \boldsymbol{\epsilon})}{T} \\ &= u_\tau \times \frac{T - r \times G(\mathbf{x}_0, \boldsymbol{\epsilon})}{T}. \end{aligned} \tag{44}$$

Considering the expectation form of $\tilde{u}_t$, there is:

$$
\begin{aligned}
\mathbb{E}_{\tilde{\mathbf{x}}_t}\left[\tilde{u}_t(\tilde{\mathbf{x}}_t|\mathbf{x}_0)\right] &= \sum p(\tilde{\mathbf{x}}_t|\mathbf{x}_0)\tilde{u}_t(\tilde{\mathbf{x}}_t|\mathbf{x}_0) \\
&= \sum p(\tilde{\mathbf{x}}_t|\mathbf{x}_0)\left[\mathbf{u}'\left(\mathbf{x}_0, \tau\right)\mathbf{x}_0 + \mathbf{v}'(\mathbf{x}_0, \tau)\boldsymbol{\epsilon}\right]\underbrace{\frac{T - r \times G(\mathbf{x}_0, \boldsymbol{\epsilon})}{T}}_{0 \leq \text{coefficient} \leq 1} \\
&\leq \sum p(\tilde{\mathbf{x}}_t|\mathbf{x}_0)\left[\mathbf{u}'\left(\mathbf{x}_0, \tau\right)\mathbf{x}_0 + \mathbf{v}'(\mathbf{x}_0, \tau)\boldsymbol{\epsilon}\right] \\
&= \mathbf{u}'\left(\mathbf{x}_0, \tau\right)\left[\sum p(\tilde{\mathbf{x}}_t|\mathbf{x}_0)\mathbf{x}_0\right] + \mathbf{v}'(\mathbf{x}_0, \tau)\boldsymbol{\epsilon} \\
&= \tilde{u}_t(\tilde{\mathbf{x}}_0|\mathbb{E}_{\tilde{\mathbf{x}}_t}[\mathbf{x}_0]).
\end{aligned}
\tag{45}
$$

Therefore, the training objective $\mathbb{E}\|\tilde{u}_t - \tilde{u}_\theta\|^2 \leq c\,\mathbb{E}\|\mathbf{x}_0 - \mathbf{x}_\theta\|^2$, where $c$ is the coefficient.

## G Code Implementations

Our framework is a module constructed on current diffusion models. We demonstrate our kernel part *rescale diffusion trajectory* with pseudo python code as below:

```python
def rescale_diffusion_trajectory(x_0, epsilon, embedding,
        labels, alphas_cumprod, timesteps, mode):
    #embedding: embedding matrix, f(x,i)=(embedding * x)[i]
    #labels: I
    #alphas_cumprod: list of all u_t
    #timesteps: t
    #mode: noising or denoising

    #1. get f(x,i):
    self_dot = torch.sum(embedding * embedding, dim=-1)
    f_x_i = self_dot[labels][..., None]
    labels = labels[..., None]

    #2. get f(x,j) and f(eps,j):
    embedding = embedding.permute(1, 0)
    f_x_j = torch.matmul(x_0, embedding)
    f_eps_j = torch.matmul(epsilon, embedding)

    #3. get f(x,i) - f(x,j): (usually >=0; smaller -> closer)
    #filter out f(x,i)-f(x,i) with a large positive number 100
    fxi_minus_fxj = (f_x_i - f_x_j).scatter(-1, labels, 100)

    #4. get f(eps,i) and f(eps,j) - f(eps,i): (larger -> more noise)
    f_eps_i = torch.gather(f_eps_j, -1, labels)
    #filter out f(eps,i)-f(eps,i) with a large negative number -100
    fepsj_minus_fepsi = (f_eps_j - f_eps_i).scatter(-1, labels, -100)

    #5. get fraction and u_t_0
    #mask results outside the support set
    info_mask = (fepsj_minus_fepsi < 0) | (fxi_minus_fxj < 0)
    fraction = fix_minus_fjx / fjeps_minus_fieps
    fraction[info_mask] = 100
    min_frac, _ = fraction.min(dim=-1) # minimum
    #Diffusion Variance Preserving eq. (9)
    u_t_0 = torch.sqrt(1 / (1 + min_frac ** 2))[..., None]

    #6. rescale timesteps
    sqrt_alphas_cumprod = torch.sqrt(alphas_cumprod)
```

```python
###!!!important trick!!!###
#We do not need to calculate the function G(x_0,t) (eq. (12)).
#Timesteps of diffusion processes are discrete and
#  we just iterate over and compare with all coefficient functions.
#Besides, function G is easy to calculate for Flow Matching.
index = torch.sum(u_t_0 < sqrt_alphas_cumprod, dim=-1)

#T is the maximum timestep, for example T=2000.
#confactor is the confidency factor
#tau is the rescaled timestep
#delta_tau is the rescaled decoding velocity
if mode == 'noising':
    tau = (timesteps + index - \
        (((timesteps + 1) / T) * index)).long().clamp(0, T)
    tau = (confactor * tau.float() + \
        (1.0 - confactor) * timesteps.float()).long().clamp(0, T)
    return tau
elif mode == 'denoising':
    delta_tau = (T - index) / T
    delta_tau = (confactor * delta_tau + \
        (1 - confactor) * 1.0).clamp(0, 1)
    return delta_tau
```

Table 6: FID of difference sampling strategies.

| | Gaussian | Deterministic |
|---|---|---|
| BINARY CODING | 13.39 | **3.86** |
| FIXED EMBEDDING | 12.21 | **9.15** |
| TRAINABLE EMBEDDING | 22.24 | **10.99** |

## H  Analysis

**Gaussian Sampling**  Our framework is compatible with the Gaussian sampling in DDPM, where random noises can be added into each iteration step. Algorithm 3 demonstrates the Gaussian sampling procedure. Compared with algorithm 2, a Gaussian noise $\mathbf{z} \sim \mathcal{N}(\mathbf{0}, \sigma_t^2 \mathbf{I})$ with a decreasing variance $\sigma_t$ is injected to the estimated next state $\tilde{\mathbf{x}}_t$. This noise $\mathbf{z}$ will be mapped as changing the initial sampling $\tilde{\mathbf{x}}_T$ through the trajectory alteration step. We illustrate the deterministic and Gaussian sampling for our model on CIFAR-10 in Table 6, where the deterministic sampling can achieve a much better performance of FID. We assume this is because our coefficient functions $\mathbf{u}(\mathbf{x}_0, t)$ and $\mathbf{v}(\mathbf{x}_0, t)$

---

**Algorithm 3** Gaussian Sampling

1: $t := T, \tau := T$
2: $\tilde{\mathbf{x}}_t \sim \mathcal{N}(\mathbf{0}, \mathbf{I})$             // Initialing
3: **for** $\Delta t := \Delta t_1, \ldots, \Delta t_s$ **do**     // $\sum_{\Delta t} = T$
4:      $\mathbf{z} \sim \mathcal{N}(\mathbf{0}, \sigma_t^2 \mathbf{I})$       // Gaussian Noise
5:      $\hat{\mathbf{x}}_0 := \mathbf{x}_\theta(\tilde{\mathbf{x}}_t, t)$        // Pseudo Target
6:      $\hat{\epsilon} := \Psi^{-1}([\tilde{\mathbf{x}}_t; \tau])$    // Trajectory Alteration
7:      $\tau_\Delta := \mathcal{T}\left(t - \Delta t, G(\hat{\mathbf{x}}_0, \hat{\epsilon})\right)$     // eq. (25)
8:      $\tilde{\mathbf{x}}_t := \mathbf{u}(\hat{\mathbf{x}}_0, \tau_\Delta)\hat{\mathbf{x}}_0 + \mathbf{v}(\hat{\mathbf{x}}_0, \tau_\Delta)\hat{\epsilon} + \mathbf{z}$
9:      $t := t - \Delta t, \tau := \tau_\Delta$      // Updating
10: **end for**
11: $\mathbf{x}_0 := \mathbf{x}_\theta(\tilde{\mathbf{x}}_t, t)$           // $\mathbf{x}_1 \rightarrow \mathbf{x}_0$
12: **return** $\mathbf{x}_0$

---

are dynamically calculated to rescale the deterministic trajectory in the training stage. In the inference stage, $\mathbf{x}_0$ is replaced by $\mathbf{x}_\theta(\mathbf{x}_t, t)$, where errors will accumulate if the predicted pseudo target changes frequently. Moreover, Gaussian sampling will further introduce random noises at each reverse step, making our rescaled timestep $\tau$ far away from the training situation. Therefore, errors in the calculations of trajectory scaling will explode over iterations.

## I  Limitations

Our framework is proposed to migrate the powerful continuous diffusion models to discrete problems. There is another technical route that directly designs the diffusion process on the discrete state space and our method is not useful for this scenario. However, we believe the continuous diffusion models can be a general framework for generative modeling and our effort can advance this target.

We prefer $\mathbf{x}_0$ as the training target because we highly depend on the reliability of the predicted $\hat{\mathbf{x}}_0$ during inference. Although it is possible to use other targets, the modeling effect will decrease in practical use, which limits the flexibility of diffusion modeling. For example, predicting the $\hat{\epsilon}$ and recovering $\hat{\mathbf{x}}_0$ with eq. (23) is inefficient, because a small error in predicting $\hat{\epsilon}$ will be amplified by eq. (23) and lead to the collapse of $G(\hat{\mathbf{x}}_0, \hat{\epsilon})$.

Our approach requires extra computational cost. But they are acceptable since our rescaling process is a series of parallel matrix computations. Considering that our approach is compatible with the Self-Conditioning [Chen et al., 2023b], our overhead is negligible when it is used.

## J  Other Experimental Details

For language modeling, we utilize the model configuration *transformer-iwslt-de-en* in FAIRSEQ framework [Ott et al., 2019] for IWSLT14 DE-EN, which has 6 transformer layers, 4 attention heads, 512 hidden dimensions, and 1024 feed forward layer dimensions. For other datasets, the configuration is *transformer-base*, which has 6 transformer layers, 8 attention heads, 512 hidden dimensions, and 2048 feed forward layer dimensions. The embedding dimension is 128. The beam size is 1 length prediction beam $\times$ 5 generation beam, since the length prediction is unstable for diffusion language models. For reranking, we take 7 length prediction beam $\times$ 3 generation beam as Difformer to let the transformer choose the best one.

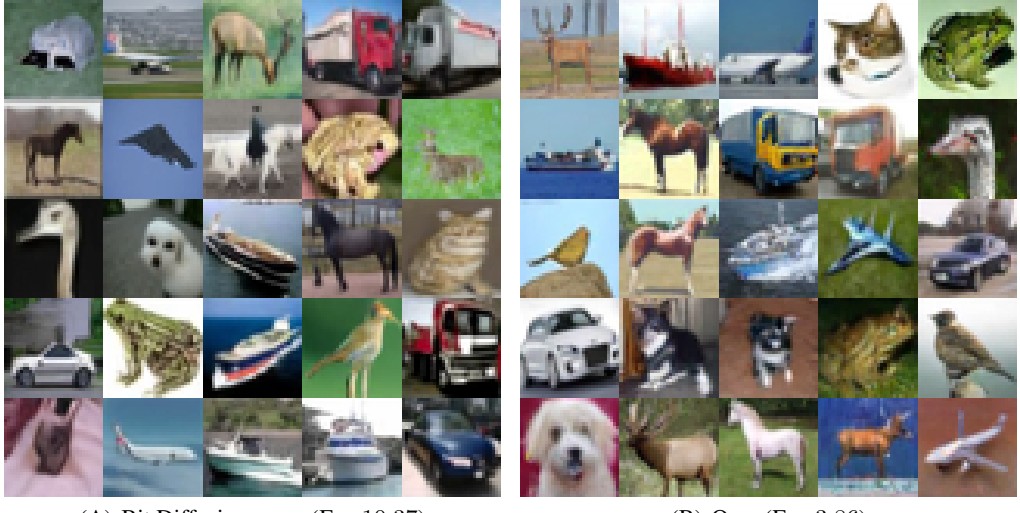

| (A) Bit Diffusion *repro* (FID 10.37) | (B) Ours (FID 3.86) |

Figure 5: Generated BINARY CODING images of reproduced Bit Diffusion and Ours on CIFAR-10.

For image generation, we set the scaling factor $r = 0.5$ for training. Besides, we find that a smaller factor for inference is sometime useful. We set $r = 0.45$ on binary coding and $r = 0.2$ on fixed embedding during inference. When the pixel embedding is learnable, the scaling factor is $r = 0.5$, which is the same as training.

Our experiments are performed with Nvidia 80G A100. Each language result requires about 2 days on one single A100. Each image result requires about a week on one single A100.

## K   Impact Statements

This paper presents work whose goal is to advance the field of Deep Learning. The datasets we used has been widely deployed for many years and has basically no negative impact. Our approach is a framework that migrates existing diffusion models to discrete problems, which does not provide a large pre-trained model that can be used to generate fake contents.

## L   Case Study

Generated sentences on IWSLT14 DE-EN and GIGAWORD are illustrated in Table 7 and Table 8. Generated images on CIFAR-10 are depicted in Figure 5, 6, and 7.

Table 7: Cases of translation on IWSLT14 DE-EN.

| Source: GERMAN | Target: ENGLISH | | |
| --- | --- | --- | --- |
| | **Difformer** | **Ours** | **Golden** |
| ich möchte ihnen erzählen , wie wir das herausgefunden haben . | i want to tell you about this . | i want to tell you how we ' ve figured that out . | i want to tell you how we found that out . |
| da gingen ganz schön viele verrückte dinge vor sich . | lots of crazy things . | there were quite a lot of crazy things going on . | there was a whole lot of crazy going on in there . |
| man macht etwas , das eigentlich ein wenig anders ist . | you do something a little different . | you ' re doing something that ' s actually a little bit different . | you do something that ' s actually a little different . |
| und die welt in der wir lebten sah so aus . | and the world we lived like this . | and the world we lived in looked like this . | and the world we used to live in looked like this . |
| man erwartet eine zusätzliche milliarde spieler im nächsten jahrzehnt . | you ' ll expect an next billion players . | you expect an extra billion players in the next decade . | they expect one billion more gamers in the next decade . |
| b hat diese vorteile und risiken . was wollen sie tun ? | b has risks . what do you want to do ? | b has these benefits and risks . what do you want to do ? | b has these benefits , and these risks . what do you want to do ? |
| wir haben also so eine situation , wo , je weiter unsere wissenschaft fortschreitet , wir uns um so mehr eingestehen müssen , dass diese kategorien , die wir für stabile anatomische kategorien gehalten hatten , welche sehr einfache zuordnungen herstellten um dauerhafte identitätskategorien zu schaffen , viel unschärfer sind , als wir angenommen haben . | so we have this situation where the continuing our science continues , we need to admit the more that these categories that we thought were stable anatomical categories , which made a very simple collaborations to create permanent identity ories are much unsharers than we ' ve assumed . | so we have a situation where , as the further our science goes on , we have to admit in terms , the more that these categories that we thought of be a stable anatomical categories , which made a very simple assaments to create permanent identity categories , are much more blanky than we ' ve accepted . | so what we have is a sort of situation where the farther our science goes , the more we have to admit to ourselves that these categories that we thought of as stable anatomical categories that mapped very simply to stable identity categories are a lot more fuzzy than we thought . |

Table 8: Cases of summarization on GIGAWORD.

| Source | Target | | |
| | **Difformer** | **Ours** | **Golden** |
| --- | --- | --- | --- |
| the asian swimming record tumbled again at the seven-day olympic test event here on friday . | asian swimming record falls again | asian swimming tumble again at olympic test event | asian swimming record tumbles again at china 's olympic trials |
| a truck carrying illegal north african immigrants flipped over in northeastern spain , killing ## and injuring six others , police said monday . | truck carrying illegal immigrants crashes in spain killing ## | ## illegal immigrants killed in truck accident in northeastern spain | ## immigrants killed in road accident in spain |
| new zealand share prices closed #.## percent lower wednesday after investors took their lead from further weakness in overseas markets , dealers said . | new zealand shares fall #.## percent | new zealand shares close #.## percent lower | new zealand shares close down #.## percent |
| the sudanese opposition said here thursday it had killed more than ### government soldiers in an ambush in the east of the country . | sudanese opposition claims over ### soldiers killed | sudanese opposition claims ### soldiers killed in ambush | sudanese opposition says ### government troops killed in ambush |
| these sports stories for release tuesday , september ## , #### , are moving today to clients of the new york times news service . | thursday 's sports budget | cox news service sports budget | cox news service tuesday sports budget |
| bangladesh and india signed a deal here thursday giving green signal to resumption of passenger train service between the two neighboring countries after ## years . | bangladesh india sign agreement on train service | bangladesh india sign agreement to resume train service | bangladesh india sign agreement for resumption of train service after ## years |

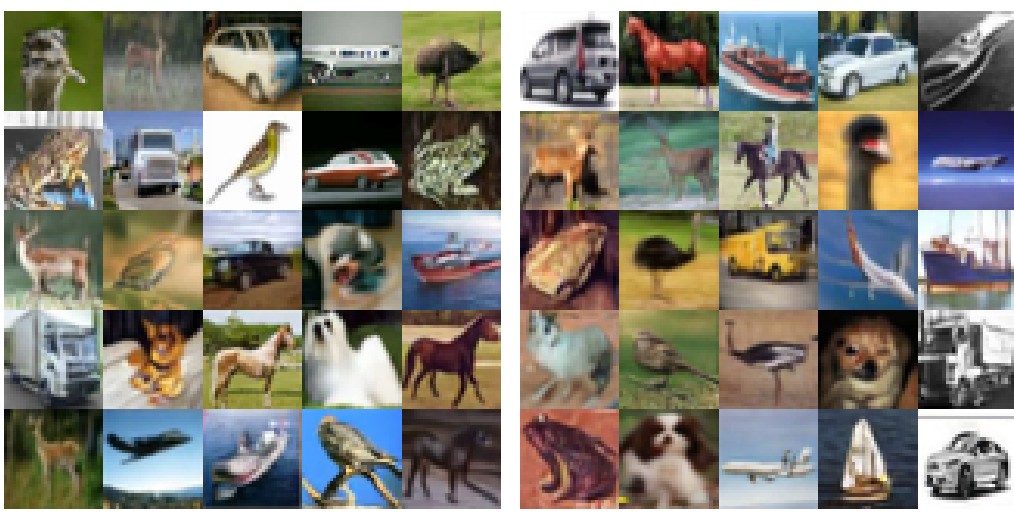

(A) Bit Diffusion *repro* (FID 12.96)   (B) Ours (FID 9.15)

Figure 6: Generated FIXED EMBEDDING images of reproduced Bit Diffusion and Ours on CIFAR-10.

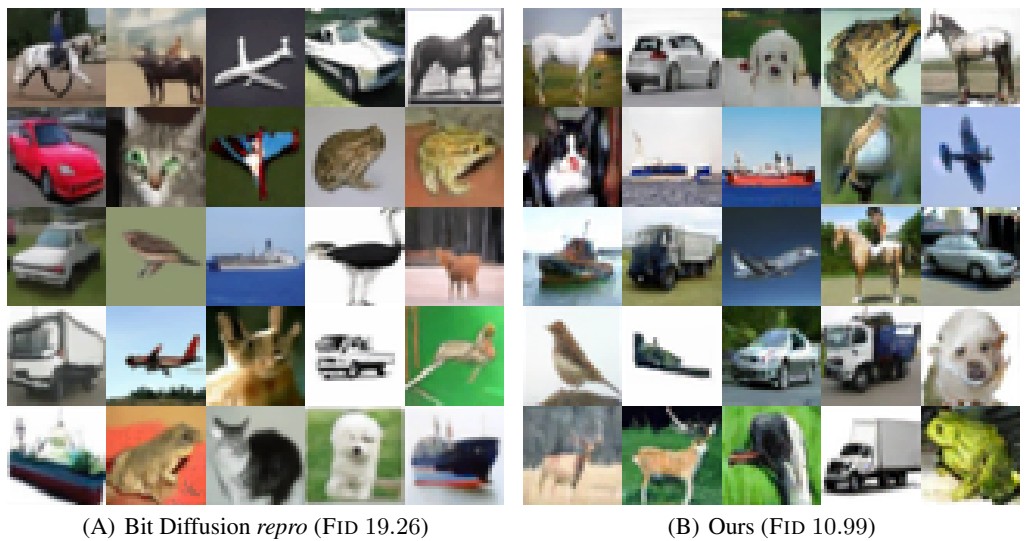

(A) Bit Diffusion *repro* (FID 19.26)          (B) Ours (FID 10.99)

Figure 7: Generated TRAINABLE EMBEDDING images of reproduced Bit Diffusion and Ours on CIFAR-10.

