# OpenReview forum: "Discrete Modeling via Boundary Conditional Diffusion Processes"
_NeurIPS.cc/2024/Conference — NeurIPS 2024 poster_

### Official Review · Reviewer_1VFK · 2024-06-30

**Soundness:** 2
**Presentation:** 2
**Contribution:** 2
**Rating:** 6
**Confidence:** 4

**Summary:**

The authors propose a discrete diffusion model that operates in the embedding space of vectors that represent discrete tokens. They propose a method that takes into account the fact that whole regions of the space will decode to the same discrete token because during decoding, the closest embedding vector is selected. The method is that at training time, instead of interpolating between $x_0$ and $x_1$, they interpolate from $x_{t_0}$ to $x_1$ where $x_{t_0}$ is the point at which the line between $x_0$ and $x_1$ intersects the 'discrete boundary' of $x_0$ which is the region of embedding space that all decodes to $x_0$. The authors test their method on language tasks as well as CIFAR-10 image modeling.

**Strengths:**

The idea of trying to exploit the fact that the embedding space has a very specific structure consisting of volumes that all decode to the same discrete token is interesting and novel.

The experimental results look quite promising with good performance compared to other discrete diffusion approaches on machine translation and text summarization.

The authors are working on a significant problem as finding diffusion methods that can perform on par with autoregressive models for discrete modeling would be very impactful.

**Weaknesses:**

The paper's main weakness is lack of clarity and precision in the description of the method. I really struggle to understand what the authors are doing exactly and how it is justified.

- It is not clear to me that your training objective and sampling regime are justified. The flow matching paradigm is to construct a $p_t(x_t | x_0)$ probability path and then find a $u_t(x_t | x_0)$ vector field that generates this probability path through the transform defined by ODE integrating along the $u_t(x_t | x_0)$ vector field. The final step is to say that a vector field that generates the unconditional probability path $p_t(x_t)$ is equal to $\mathbb{E}\_{p(x_0 | x_t)} [ u_t(x_t | x_0) ]$ and this can be learned tractably with an objective of $\mathbb{E}\_{p(x_0) p(x_t | x_0)} [ || v_t^\theta(x_t) - u_t(x_t | x_0) ||^2 ]$. However, I don't see how your method fits in this framework. This is for two reasons: 1) at no point do you define the conditional vector field $\tilde{u}\_t(\tilde{x}\_t | x\_0)$ that generates your rescaled probability path $\tilde{p}\_t(\tilde{x}\_t | x\_0)$. How are you able to generate along these probability paths without defining $\tilde{u}\_t(\tilde{x}\_t | x\_0)$ ? 2) You sample using $u\_t(x\_t | x\_0)$ for the original $p\_t(x\_t | x\_0)$ but with the $\hat{x}\_0$ prediction from the neural network. This is valid in the original flow matching case because $u\_t(x\_t | x\_0)$ is linear and so the $\mathbb{E}\_{p(x\_0) p(x\_t | x\_0)} [ || v\_t^\theta(x\_t) - u\_t(x\_t | x\_0) ||^2 ]$ objective can be re-arranged into $x\_0$ prediction $\mathbb{E}\_{p(x\_0) p(x\_t | x\_0)} [ || x\_\theta(x\_t , t) - x\_0 ||^2 ]$. However we don't necessarily know if $\tilde{u}\_t(\tilde{x}\_t | x\_0)$ that generates $\tilde{p}\_t(\tilde{x}\_t | x\_0)$ is linear so we can't be sure it is valid to re-arrange to $x\_0$ prediction.
- I think after the method is explained in section 3, the authors should have text that revisits the probability contour intuition they provide during the introduction. I think this is a nice mental model to understand the method but it is very unclear how exactly the method presented in section 3 gives you the desired probability contours that are promised in the introduction. For example, if a contour is very long in embedding space, then definitely the far away points on the contour with have low probability under $\tilde{p}\_t(\tilde{x}\_t | x\_0)$ so therefore probability is not constant along the discrete boundary and therefore the discrete boundary does not line up with the probability contour. These kinds of doubts would be alleviated with a clearer presentation linking back to the original probability contour view and some more precise statements.
- The discussion relating to the confidence factor is very unclear. Firstly, I'm unsure why it is even called a confidence factor, we can be fully confident that our data is discrete so why is there uncertainty about this? It is also concerning that the method can oscillate and mode collapse when r=1 which is your full method. This sounds like a major flaw of the approach if it cannot work at all when your new method is used in isolation. This warrants more discussion regarding this failure case. Furthermore L195 should be explained more as it is very unclear what "fixing the initial path" means and it sounds very important if it is required for your method to work.


Regarding the experiments:
- It seems unfair to allow your method to be re-ranked in Table 1 if you don't also re-rank the transformer baseline
- In Table 3, it is hard to get a good sense of how your method compares to other discrete diffusion approaches. Since they all use different embedding strategies it seems that it is only a good comparison to compare methods within the same embedding strategy. In this case, you are really only comparing to bit diffusion but your reproduction seems to perform much worse than the originally stated results in the bit diffusion paper. It is therefore difficult to place your method with confidence amongst the other discrete diffusion methods.



My score is given on the basis of the very unclear presentation meaning I cannot understand the method the author's are proposing. If this can be cleared up in the rebuttal period, I am happy to raise my score.

**Questions:**

Here I have listed some more minor questions for the authors that can help improve the exposition. My main concerns are given in the previous discussion.

- How does your formulation handle the fact that for some $x\_0$ and $\epsilon$ samples, there will never be a boundary crossing because $x\_0$ may be on the outside of the space or $\epsilon$ is sampled within $x\_0$'s discrete boudary.
- On L187 (24) is already deterministic because its an ODE so I don't understand why you then propose a deterministic alternative.
- In Table 2 it is unclear what ablation you are actually doing, what does it mean to only rescale the forward process versus rescaling the forward and backward?

**Limitations:**

I don't believe the authors adequately describe the limitations of their method when r is near 1, they say that it can be unstable but provide no further details or explanation regarding this failure case.

---

> ### Author Rebuttal · Authors · 2024-08-06
>
> We sincerely appreciate your meticulous review and the valuable comments. Please kindly find our response below.
>
> **Q1: The rescaled vector field and probability path**
>
> *Due to the character limit, this is only an outline. Please refer to comments for more details.*
>
> 1. Our neural network predicts $\mathbf{x}_0$ not $\tilde{u}_t(\tilde{\mathbf{x}}_t|\mathbf{x}_0)$.
>
> 2. $\tilde{u}_t(\tilde{\mathbf{x}}_t|\mathbf{x}_0)$ is tractable but we did not derive it because it is not used.
>
> 3. We generate the probability path with the deterministic reverse diffusion process (Equation 25).
>
> 4. The training objective $||\mathbf{x}_0-\mathbf{x} _\theta||$ is valid. Derivations in Comments.
>
> **Q2: Method and the probability contour**
>
> *Due to the character limit, this is only an outline. Please refer to comments for more details.*
>
> 1. We have demonstrated the rescaled probability contours of $\tilde{p}_t(\tilde{\mathbf{x}}_t|\mathbf{x}_0)$ in Figure 2, where sample points are exactly calculated by our proposed method (Equation 22).
>
> 2. $\tilde{p}_t(\mathbf{x}|\mathbf{x}_0)$ is a series of functions, where the consistency to the boundary contour is inversely proportional to the subscript $t$.
>
> **Q3: Confidence factor**
>
> *Due to the character limit, this is only an outline. Please refer to comments for more details.*
>
> 1. The confidence factor is named after some thoughts of sampling, such as the confidence interval. Our data is discrete and the uncertainty comes from the learned diffusion model.
>
> 2. The mode collapse when $r=1$ comes from two part: One is the uncertainty of the learned diffusion model as described above. The other involves a traditional problem of diffusion processes.
>
> 3. Fixing the initial path means the Trajectory Alteration (Line 8 in Algorithm2) is optional in practice. This is a negligible detail of our model.
>
> **Q4: Re-ranking**
>
> This is a fair comparision because beam searching the generated results of the transformer is re-ranking with transformer.
>
> Re-ranking is a traditional strategy widely used in Non-autoregressive translation[7,8]. We use the generated results of our method re-ranked by the transformer to demonstrate their upper bound. Since the generation of diffusion LMs are highly random, their own probability scores often cannot reflect the upperbound of their performance. Re-ranking the generated results with an auto-regressive model can better demonstrate the potential capability of the diffusion model.
> Besides, re-ranking the transformer baseline just gets what it generates. Because **beam search is exactly re-ranking the generated contents with the transformer's probabilities themselves.**
>
> **Q5: Table 3**
>
> 1. **We believe it is a reasonable comparision because we strictly follows the setting of BitDiffusion[9] as in the Table 1 of their paper.**
> We want to clarify that the embedding strategy is only valid for continuous diffusion models, where there is not embedding stage for discrete diffusion models.
> We compare the continuous diffusion models and the discrete diffusion models under the same level of continuous information, Continuous Pixels, Discrete Ordinal Pixels, and Categorical Pixels, where the latter the less continuous information models can obtain.
>
> 2. **We have clarified in Lines 285-287 that since they only provide the model code without a training script with detailed hyperparameters or a checkpoint, we have to reproduce with exactly the same configuration based on their code and paper.** We have done our best effort to reproduce the results and we promise to release our code and reproduction. Besides, we achieve the best reproduction result in the open source community.  We are confident in the reliability of our experimental results and hope to address the reviewer's concerns.
>
> 3. **Even ignoring the above issues, our method still shows effectiveness.** Considering the gap introduced by the continuous information that the original BitDiffusion (Fid 3.48) can not beat DDPM (Fid 3.17) under the Gaussian sampling, our method (Fid 3.86) achieves improvement over DDIM (FiD 4.04), which are both deterministic diffusion processes. This means that our approach can achieve better results with less continuous information.
>
> **Q6: Samples in the boundary**
>
> 1. **It is possible that a random noise $\boldsymbol{\epsilon}$ is sampled within $\mathbf{x}_0$'s discrete boundary.** However, the probability of this happening in a high-dimensional representation space is very low. According to our statistics during the training phase, this probability is less than 0.1%.
>
> 2. **Our solution is demonstrated in the pseudo code (Line 639, Appendix F), where we mask out this situation when $f(\boldsymbol{\epsilon},\mathcal{I})>f(\boldsymbol{\epsilon},\mathcal{J})$.** This means that random noise $\boldsymbol{\epsilon}$ sampled within $\mathbf{x}_0$'s discrete boundary will not be rescaled and its trajectory is the same as original DDPM.
>
> **Q7: Line 187**
>
> Similar to 3.2 in Q1, as demonstrated in Lines 186-189, we provide an alternative for the ODE decoding, which is a deterministic reverse process. **The word 'deterministic' is used to reflect the difference between our approach and traditional Gaussian process in DDPM**, because both of them model the reverse process.
>
> **Q8: Ablation in Table 2**
>
> 1. **First three lines in Table 2 reveal where the performance improvement of our method mainly comes from.**
> Only rescale the forward process means we just train the diffusion model with the rescaled trajectory and reverse the trajectory as in the DDPM. Rescaling the forward and backward means we use the rescaled trajectory for both training and inference. The results in Table 2 shows that training with the rescaled trajectories makes a major contribution to the final performance of our method.
>
> 2. **Last line in Table 2 reveals that our method is robust for different trajectory types that optimal transport trajectory used in Flow Matching can sometimes achieve better results.**

---

> > ### Comment · Reviewer_1VFK · 2024-08-08
> > **Response to Authors**
> >
> > Thank you for the detailed rebuttal. I am happy with the responses to my questions regarding the probability contour, confidence factor, samples on the boundary, the deterministic sampler and Table 2.
> >
> > However, I am still confused about your overall method. I appreciate the derivation of $\tilde{u}_t(\tilde{x}_t | x_0)$ however this still leaves me with questions about the loss you use. You justify the $x_0$ prediction by saying that we are trying to match $\tilde{u}_t(\tilde{x}_t | x_0)$ with $\tilde{u}_t(\tilde{x}_t | x_0^\theta)$ which yes they would be equal if you set $x_0 = x_0^\theta$. However this misunderstands what the loss is actually doing. When we do flow matching, we use the L2 loss because we would like to have $\tilde{u}_t^\theta(\tilde{x}_t) = \mathbb{E}[ \tilde{u}_t(\tilde{x}_t | x_0) | \tilde{x}_t] $ which is the solution to min $\mathbb{E} [ || \tilde{u}_t(\tilde{x}_t | x_0) - \tilde{u}_t^\theta(\tilde{x}_t) ||^2 ]$. If you then solve min $\mathbb{E} [ || x_0 - x_0^\theta (\tilde{x}_t) ||^2]$ you will obtain $\mathbb{E} [ x_0 | \tilde{x}_t ]$. But we don't necessarily have $ \mathbb{E}[ \tilde{u}_t(\tilde{x}_t | x_0) | \tilde{x}_t] = \tilde{u}_t(\tilde{x}_t | \mathbb{E} [ x_0 | \tilde{x}_t ]) $ which would be required for your loss to be justified. This is ok in standard flow matching since $u$ is linear but it is not clear in your case.
> >
> > Regarding Table 1, am I to understand you used beam-search for the transformer baseline and this is why you say this is equivalent to re-ranking your method? Why didn't you just use normal sampling for the transformer baseline and normal sampling for your method, I feel like that would be a much clearer narrative and not have to imply your method requires re-ranking to work well.
> >
> > For Table 3 it is good that you implemented Bit Diffusion and got the best open source results, however, it doesn't really help make your narrative clear since the results are worse than those reported in the original bit diffusion paper. Why didn't you just use standard flow matching on the same discrete embedding space as your method but just without your boundary method. That would have been the closest and simplest baseline and would have made the precise benefits of your method clear.

---

> ### Author Response · Authors · 2024-08-06
> **Detailed answer for Q1: The rescaled vector field and probability path (Part 1/2)**
>
> **Q1: The rescaled vector field and probability path**
>
> We do not define the vector field $\tilde{u}_t(\tilde{\mathbf{x}}_t|x_0)$ in our framework, because our neural network predicts $\mathbf{x}_0$ and we generate the probability path with the deterministic reverse diffusion process.
>
> 1. **Why do we predict $\mathbf{x}_0$?**
>
> $G(\mathbf{x}_0,\boldsymbol{\epsilon})$ is the key of our method during both training and inference stage, we are required to have a low error estimation on $\mathbf{x}_0$ (Lines 180-182). Predicting $\tilde{u}_t(\tilde{\mathbf{x}}_t|x_0)$ or $\boldsymbol{\epsilon}$ will amplify the errors in the neural network's output.
>
> 2. **How to define the vector field $\tilde{u}_t(\tilde{\mathbf{x}}_t|\mathbf{x}_0)$ if we want?**
>
> > Preliminary: $u_t(\mathbf{x}_t|\mathbf{x}_0) = \frac{\mathrm{d}\mathbf{x}_t}{\mathrm{d}t}$ (Equation 11-13 in Flow Matching [1])
> >
> > In our framework: $\tilde{\mathbf{x}}_t = \mathbf{u}(\mathbf{x}_0,\mathcal{T}(t,G(\mathbf{x}_0,\boldsymbol{\epsilon})))\mathbf{x}_0+\mathbf{v}(\mathbf{x}_0,\mathcal{T}(t,G(\mathbf{x}_0,\boldsymbol{\epsilon})))\boldsymbol{\epsilon}$  (Equation 22)
> >
> > Therefore: $\tilde{u}_t(\tilde{\mathbf{x}}_t|\mathbf{x}_0) = \frac{\mathrm{d}\tilde{\mathbf{x}}_t}{\mathrm{d}t} = \left[\mathbf{u}'(\mathbf{x}_0,\mathcal{T}(t,G(\mathbf{x}_0,\boldsymbol{\epsilon})))\mathbf{x}_0+ \mathbf{v}'(\mathbf{x}_0,\mathcal{T}(t,G(\mathbf{x}_0,\boldsymbol{\epsilon})))\boldsymbol{\epsilon}\right]\frac{\mathrm{d}\mathcal{T}(t,G(\mathbf{x}_0,\boldsymbol{\epsilon}))}{\mathrm{d}t}$
> >
> > Given: $\mathcal{T}(t,t_0) = r\times t_0 + t\times (T-r\times t_0)/T$ (Eqaution 19), we have $\frac{\mathrm{d}\mathcal{T}(t,G(\mathbf{x}_0,\boldsymbol{\epsilon}))}{\mathrm{d}t}=\frac{T-r\times G(\mathbf{x}_0,\boldsymbol{\epsilon})}{T}$
> >
> > Let $\tau$ denote $\mathcal{T}(t,G(\mathbf{x}_0,\boldsymbol{\epsilon}))$
> >
> > Hence:  $\tilde{u}_t(\tilde{\mathbf{x}}_t|\mathbf{x}_0) = \frac{\mathrm{d}\tilde{\mathbf{x}}_t}{\mathrm{d}t} = \left[\mathbf{u}'(\mathbf{x}_0,\tau)\mathbf{x}_0+ \mathbf{v}'(\mathbf{x}_0,\tau)\boldsymbol{\epsilon}\right]\frac{T-r\times G(\mathbf{x}_0,\boldsymbol{\epsilon})}{T}$, which is actually tractable.
>
> 3. **How to generate the probability paths?**
>
> - 3.1 **One direct solution is Equation 24, which is derived from the Theorem3 in Flow Matching[1]**
>
> > Let $\tilde{p}_t(\mathbf{x}|\mathbf{x}_0)$ be a probability path, given equation 22, there is $\boldsymbol{\epsilon}=\frac{\mathbf{x}-\mathbf{u}(\mathbf{x}_0,\tau)\mathbf{x}_0}{\mathbf{v}(\mathbf{x}_0,\tau)}$.
> >
> > Replacing $\boldsymbol{\epsilon}$ in $\tilde{u}_t(\tilde{\mathbf{x}}_t|\mathbf{x}_0)$ with $\mathbf{x}$, we have $\tilde{u}_t(\mathbf{x}|\mathbf{x}_0)= \left[ \frac{\mathbf{v}'(\mathbf{x}_0,\tau)}{\mathbf{v}(\mathbf{x}_0,\tau)}(\mathbf{x}-\mathbf{u}(\mathbf{x}_0,\tau)\mathbf{x}_0)+\mathbf{u}'(\mathbf{x}_0,\tau)\mathbf{x}_0\right]\frac{T-r\times G(\mathbf{x}_0,\boldsymbol{\epsilon})}{T}$.
> >
> > Therefore, we generate the probability path with $\tilde{u}_t(\mathbf{x}|\mathbf{x}_0)$.
>
> However, **this may be inefficient in practical use because we have to solve the equation $\tau=\mathcal{T}(t,G(\mathbf{x}_0,\frac{\mathbf{x}-\mathbf{u}(\mathbf{x}_0,\tau)\mathbf{x}_0}{\mathbf{v}(\mathbf{x}_0,\tau)}))$ to get the $\tau$ with respect to the change of $\mathbf{x}$ in real time.** This is tractable for special cases such as original Flow Matching (Equation 42 in Appendix E), but it will be much more complex for Diffusion trajectories (Equation 43 in Appendix E). Since our approach is a general framework for all continuous diffusion processes, including both Diffusion Model and Flow Matching, **we actually use an alternative approach similar to the DDIM[2]**.
>
> *It's worth noting that we just demonstrate the vector field function $u_t(\mathbf{x}|\mathbf{x}_0)$ in Equation 24 to show its complexity. We do not provide the derivation of $\tilde{u}_t(\mathbf{x}|\mathbf{x}_0)$ because we do not want to distract readers from our actual generation method by spending too much time introducing and deriving this function that is totally not used. However, as pointed in this question by reviewer, we will add the above derivation into the appendix to ensure the integrity our framework.*

---

> ### Author Response · Authors · 2024-08-06
> **Detailed answer for Q1: The rescaled vector field and probability path (Part 2/2)**
>
> - 3.2 **Our alternative solution is Equation 25 and Algorithm 2, which works like the DDIM[2].**
>
> In short, our solution can be understood as deterministic reverse diffusion process or ODE with discrete time steps. We use a state transfer probability (Equation 25) $\tilde{p}([\tilde{\mathbf{x}}_ {t_ 1};\tau_ 1]|[\tilde{\mathbf{x}}_ {t_ 2};\tau_2],\mathbf{x}_ 0)$ to extend the traditional reverse process $p(\mathbf{x}_ {t_ 1}|\mathbf{x}_ {t_ 2},\mathbf{x}_ 0)$, where $1\leq t_1 < t_2 \leq T$. As in diffusion processes[3], if we set the timestep interval as 1, the probability path is obtained with $p_ t(\mathbf{x}|\mathbf{x}_ 0) = p(\mathbf{x}_ T|\mathbf{x}_ 0) \prod\limits_{s=t+1}^{T} p(\mathbf{x}_ {s-1}|\mathbf{x}_ {s},\mathbf{x}_ 0)$.
>
> In our framework, we get $\tilde{p}_ t([\tilde{\mathbf{x}};\tau]|\mathbf{x}_ 0)=p([\tilde{\mathbf{x}}_ T;T]|\mathbf{x}_ 0)\prod\tilde{p}([\tilde{\mathbf{x}}_ {t_ i};\tau_ i]|[\tilde{\mathbf{x}}_ {t_ {i+1}};\tau_ {i+1}],\mathbf{x}_ 0)$.
>
> Since the state transfer probability is a Dirac delta function, there is no randomness in the reverse process, where the $[\tilde{\mathbf{x}}_t;\tau]$ pair can be iteratively generated deterministically (Algorithm 2).
>
> 4. **How to derive the training objective $\min ||\mathbf{x}_ 0-\mathbf{x}_ \theta||$?**
>
> - 4.1 **From the perspective of the deterministic reverse process**
>
> We utilize the deterministic reverse diffusion process to generate data (Equation 25). Similar to DDPM[3], the learning objective for the unrescaled deterministic reverse diffusion process is derived in Appendix C.3 Equation 38, which can still be simplified to $||\mathbf{x}_ 0-\mathbf{x}_ \theta||$. For our rescaled reverse process $\tilde{p}([\tilde{\mathbf{x}}_ {t_\Delta};\tau_ \Delta]|[\tilde{\mathbf{x}}_ {t};\tau],\mathbf{x}_ 0)$, where both $\tau$ and $\tilde{\mathbf{x}}_ t$ are inputs, the objective is an equation set:
>
> $$
> \left\[\begin{aligned}
> \mathbf{u}_ {\tau_ \Delta}\mathbf{x}_ 0 + \mathbf{v}_ {\tau_ \Delta}\hat{\boldsymbol{\epsilon}}&= \mathbf{u}_ {\tau_ \Delta} \mathbf{x}_ \theta + \mathbf{v}_ {\tau_ \Delta}\hat{\boldsymbol{\epsilon}}\\\\
> \mathcal{T}(t-\Delta t,G(\mathbf{x}_ 0,\hat{\boldsymbol{\epsilon}})) &= \mathcal{T}(t-\Delta t,G(\mathbf{x}_ \theta,\hat{\boldsymbol{\epsilon}}))
> \end{aligned}\right\] \Rightarrow \left\[\begin{aligned}
> \mathbf{x}_ 0 &=\mathbf{x}_ \theta \\\\
> G(\mathbf{x}_ 0,\hat{\boldsymbol{\epsilon}}) &= G(\mathbf{x}_ \theta,\hat{\boldsymbol{\epsilon}})
> \end{aligned}\right\],
> $$
>
> where $\hat{\boldsymbol{\epsilon}}=\Psi^{-1}([\tilde{\mathbf{x}}_ t,\tau])$ is a constant.
> Besides, $\tau_ \Delta$ is a constant for the above equation.
>
> The unique solution to this equation set is $\mathbf{x}_\theta=\mathbf{x}_0$.
>
> * 4.2 **From the perspective of re-arranging**
>
> Given $\tilde{u}_ t(\tilde{\mathbf{x}}_ t|\mathbf{x}_ 0) = \left[\mathbf{u}'(\mathbf{x}_ 0,\mathcal{T}(t,G(\mathbf{x}_ 0,\boldsymbol{\epsilon})))\mathbf{x}_ 0+ \mathbf{v}'(\mathbf{x}_ 0,\mathcal{T}(t,G(\mathbf{x}_ 0,\boldsymbol{\epsilon})))\boldsymbol{\epsilon}\right]\frac{T-r\times G(\mathbf{x}_ 0,\boldsymbol{\epsilon})}{T}$, we can define $\tilde{v}_ \theta(\tilde{\mathbf{x}}_ t)=\tilde{u}_ t(\tilde{\mathbf{x}}_ t|\mathbf{x}_ \theta)$. To get the $\min ||\tilde{u}_ t(\tilde{\mathbf{x}}_ t|\mathbf{x}_ 0) - \tilde{u}_ t(\tilde{\mathbf{x}}_ t|\mathbf{x}_ \theta)||$, we can direct calculate the partial derivate. For any $i$ dimension of $\mathbf{x}_\theta$, there is:
>
> $$
> \frac{\partial ||\tilde{u}_ t(\tilde{\mathbf{x}}_ t|\mathbf{x}_ 0) - \tilde{u}_ t(\tilde{\mathbf{x}}_ t|\mathbf{x}_ \theta) ||}{\partial \mathbf{x}^i_ \theta} = \frac{\tilde{u}_ t(\tilde{\mathbf{x}}_ t|\mathbf{x}_ 0) - \tilde{u}_ t(\tilde{\mathbf{x}}_ t|\mathbf{x}_ \theta)}{||\tilde{u}_ t(\tilde{\mathbf{x}}_ t|\mathbf{x}_ 0) - \tilde{u}_ t(\tilde{\mathbf{x}}_ t|\mathbf{x}_ \theta) || } \frac{\partial \tilde{u}_ t(\tilde{\mathbf{x}}_ t|\mathbf{x}_ \theta) }{\partial \mathbf{x}^i_ \theta} =0
> $$
>
> We can drop $\frac{\partial \tilde{u}_ t(\tilde{\mathbf{x}}_ t|\mathbf{x}_ \theta) }{\partial \mathbf{x}^i_ \theta}$, because it can not provide valid value of $\mathbf{x}_ \theta$ without the term of $\mathbf{x}_ 0$.
>
> Solving the equation $\tilde{u}_ t(\tilde{\mathbf{x}}_ t|\mathbf{x}_ 0) - \tilde{u}_ t(\tilde{\mathbf{x}}_ t|\mathbf{x}_ \theta) = 0$ is difficult, but it is easy to prove that $\mathbf{x}_ \theta = \mathbf{x}_ 0$ is a solution.
>
> Since $\tilde{u}_ t(\tilde{\mathbf{x}}_ t|\mathbf{x}_ 0)$ is deterministic, it is an injection function. This means that the same input will not produce different outputs. Therefore, $\mathbf{x}_ \theta = \mathbf{x}_ 0$ can always get the minimum value of $||\tilde{u}_ t(\tilde{\mathbf{x}}_ t|\mathbf{x}_ 0) - \tilde{u}_ t(\tilde{\mathbf{x}}_ t|\mathbf{x}_ \theta) ||$ and we simplify the objective to $\min ||\mathbf{x}_ 0 - \mathbf{x}_ \theta||$ for convenience and training stability.

---

> ### Author Response · Authors · 2024-08-06
> **Detailed answer for Q2: Method and the probability contour**
>
> **Q2: Method and the probability contour**
>
> 1. **We have demonstrated the rescaled probability contours of $\tilde{p}_t(\tilde{\mathbf{x}}_t|\mathbf{x}_0)$ in Figure 2, where sample points are exactly calculated by our proposed method (Equation 22).**
>
> As illustrated in Figure 2, the probability contours calculated by our method can be easily adapted to different discrete boundaries and noising trajectories, which are inline with our expectations in Figure 1 of the Introduction Section.
>
> *Thank you for this suggestion and we will add a quick revision at the end of the Method Section and link our formula to the Figure 2.*
>
> 2. **$\tilde{p}_t(\mathbf{x}|\mathbf{x}_0)$ is a series of functions, where the consistency to the boundary contour is inversely proportional to the subscript $t$.**
>
> As illustrated in Figure 2 and Equations 21 and 22, $\tilde{p}_0(\mathbf{x}|\mathbf{x}_0)$ is exactly the boundary contour while $\tilde{p}_T(\mathbf{x}|\mathbf{x}_0)$ is a Gaussian distribution. When a boundary contour is very long in embedding space, far away point will gradually have a lower probability density under $\tilde{p}_t(\mathbf{x}|\mathbf{x}_0)$ as $t$ increases. This means, **during the inference stage, the density probability will gradually be consistent with the contour when approaching the boundary, which is inline with Lines 46-51 and Figure 1B.**
>
> Besides, we will not face the extreme case where the boundary contour is an infinitely long straight line in practical application. Because the embedding space is a finite space and the values of the embedding points will be normalized into a finite range. Therefore, when we set the diffusion space a bit larger than the embedding space, there are always series of probability functions $\tilde{p}_t(\mathbf{x}|\mathbf{x}_0)$ from Gaussian distributions to the boundaries.

---

> ### Author Response · Authors · 2024-08-06
> **Detailed answer for Q3: Confidence factor**
>
> **Q3: Confidence factor**
>
> 1. **The confidence factor is named after some thoughts of sampling, such as the confidence interval. Our data is discrete and the uncertainty comes from the learned diffusion model.**
>
> Consider a simplified situation of only two tokens 'A' and 'B', which is the binary classification. When the confidence factor $r=1$, points $\mathbf{x}$ on the boundary of 'A' strictly follow $p(\mathbf{x}\in A)=p(\mathbf{x} \in B) = 0.5$. Since the learned diffusion model is expected to guide a random noise to this boundary, any subtal perturbation to the output of our learned diffuison model may lead to a reverse of the $\mathbf{x}$'s attribution. If we decrease the factor $r$ so that points on the boundary of the token 'A' have higher probability density, e.g., $p(\mathbf{x}\in A)=0.7>p(\mathbf{x}\in B)$, we will be more confident that our generated points of the learned diffusion model belong to the token 'A'. (In this situation, $p(\mathbf{x}\in A)\geq 0.7$ is the discrete area of A, $p(\mathbf{x}\in B)\geq 0.7$ is the discrete area of B, and $0.3 < p(\mathbf{x}\in A)< 0.7$ is the unattributed area.)
>
> 2. **The mode collapse when $r=1$ comes from two part: One is the uncertainty of the learned diffusion model as described above. The other involves a traditional problem of diffusion processes.**
>
> - 2.1 **From the perspective of uncertainty**
>
> Facing tasks with strong conditions, we are confident with the model's prediction since this is a easy task. Therefore, $r=1$ works perfect (Lines 227-229 and Table 1) as the strong conditions greatly reduce uncertainty of the learned model during inference.
>
> When the condition is weak or there is no condition, uncertainty increases rapidly. This means that the sampled noisy training data constitutes a more difficult task with $r=1$. When the task difficulty exceeds the model's capability, the collapse occurs.
>
> - 2.2 **From the perspective of diffusion processes**
>
> This is also a problem similar to the well studied one in Flow Matching with Dynamic Optimal Transport [4,5,6], i.e., the path from the noise to target is too long and learning this path is hard. For example, suppose there are two unconditional targets 'A' and 'B', the sampled noisy data $\tilde{\mathbf{x}}_A$ of 'A' when $r=1$ may have a shorter path to 'B' than 'A'. Likewise, $\tilde{\mathbf{x}}_B$ can be closer to A. The diffusion models are required to map $\tilde{\mathbf{x}}_A$ to 'A' and $\tilde{\mathbf{x}}_B$ to 'B', but it is easier to learn the mapping of $\tilde{\mathbf{x}}_A$ to 'B'. **This is a general problem for all diffusion models, not just ours.** We believe the stability and performance of our method can be further improved with the Dynamic Optimal Transport, but we want to have a fair comparision with our baselines to demonstrate our effectiveness. Therefore, we just use a smaller $r$ to stabilize the training process and use the sub-optimal results to compare with the baselines.
>
> Besides, the failure case is just the unexpected loss value during training, e.g., NaN, where these models are unable to generate contents.
>
> 3. **Fixing the initial path means the Trajectory Alteration (Line 8 in Algorithm2) is optional in practice. This is a negligible detail of our model.**
>
> - 3.1 **From the perspective of uncertainty**
>
> Line 8 in Algorithm 2 is to update the trajectory. If the learned diffusion model predicts a different $\mathbf{x}_0$, we have to update the original noise because currently we are in a different trajectory. If the learned model keeps changing the predicted target, which means it is uncertain about where to go, updating the trajectory correspond to an unreliable target may not be beneficial for the denoising process. If the learned model rarely changes its prediction, it means we are on the right path and no updates are needed. Therefore, it would be a viable option to discard this step (Line 8 in Algorithm 2) to make decoding faster.
>
> - 3.2 **From the perspective of experiments**
>
> As illustrated in Table 3, if we train with the rescaled forward process and decode with unrescaled trajectory, there will be only a slight performance degradation. The performance improvement of our model mainly comes from the training process, and **some small changes in the decoding process may not have much impact.**
> In practical application, we just discard the trajectory update (Line 8 in Algorithm 2) for simplicity in our experiments and find almost no performance degradation.

---

> ### Author Response · Authors · 2024-08-06
> **References in Rebuttal**
>
> [1] Flow Matching for Generative Modeling
>
> [2] Denoising Diffusion Implicit Models
>
> [3] Denoising Diffusion Probabilistic Models
>
> [4] Flow Straight and Fast: Learning to Generate and Transfer Data with Rectified Flow
>
> [5] Rectified Flow: A Marginal Preserving Approach to Optimal Transport
>
> [6] Improving and Generalizing Flow-Based Generative Models with Minibatch Optimal Transport
>
> [7] Non-Autoregressive Neural Machine Translation
>
> [8] Understanding Knowledge Distillation in Non-Autoregressive Machine Translation
>
> [9] Analog Bits: Generating Discrete Data Using Diffusion Models with Self-Conditioning

---

> ### Author Response · Authors · 2024-08-09
> **Response to Reviewer**
>
> We sincerely appreciate your response, and we are delighted to have addressed some of your concerns. We hope to better align our contributions with your understanding.
>
> **Q1: loss function**
>
> $||\mathbf{x}_ 0 - \mathbf{x}_ \theta||$ is a simplified objective where we drop the coefficent (Equation 38 in Appendix C.3).
>
> Let's rewrite the expectation $\mathbb{E}_{\tilde{\mathbf{x}}_t}[\tilde{u}_t(\tilde{\mathbf{x}}_t|\mathbf{x}_0)]$  in the sum form, we have:
>
> $$
> \underbrace{\sum p(\tilde{\mathbf{x}}_ t) \tilde{u}_ t(\tilde{\mathbf{x}}_ t|\mathbf{x}_ 0)}_ {\mathbb{E}_ {\tilde{\mathbf{x}}_ t}[\tilde{u}_ t(\tilde{\mathbf{x}}_ t|\mathbf{x}_ 0)]} = \sum p(\tilde{\mathbf{x}}_ t) \left[\mathbf{u}'\mathbf{x}_ 0+ \mathbf{v}'\boldsymbol{\epsilon}\right]\underbrace{\frac{T-r\times G(\mathbf{x}_ 0,\boldsymbol{\epsilon})}{T}}_ {0<\text{coefficient}<1} \leq \sum p(\tilde{\mathbf{x}}_ t) \left[\mathbf{u}'\mathbf{x}_ 0+ \mathbf{v}'\boldsymbol{\epsilon}\right] = \underbrace{\left[\mathbf{u}' \sum p(\tilde{\mathbf{x}}_ t) \mathbf{x}_ 0 + \mathbf{v}'\boldsymbol{\epsilon}\right]}_ {\tilde{u}_ t(\tilde{\mathbf{x}}_ t|\mathbb{E}_ {\tilde{\mathbf{x}}_ t}[\mathbf{x}_ 0])},
> $$
> where $\mathbf{x}_ 0$ is the function of $\tilde{\mathbf{x}}_ t$ and $\frac{T-r\times G(\mathbf{x}_ 0,\boldsymbol{\epsilon})}{T}$ is a dynamic coefficient ranging from $0$ to $1$. Our current objective minimizes the upperbound of $\min \mathbb{E}||\tilde{u}_t - \tilde{u}^\theta_t||$, which is also effective.
>
> As illustrated in Equation 38 of Appendix C.3, we simplify the training objective and drop the coefficient during training, which is similar to the equation 14 in DDPM. This coefficient works as to dynamically assign weights to different samples based on the noise and x0. Therefore, if we want to make $\mathbb{E}_ {\tilde{\mathbf{x}}_ t}[\tilde{u}_ t(\tilde{\mathbf{x}}_ t|\mathbf{x}_ 0)]=\tilde{u}_ t(\tilde{\mathbf{x}}_ t|\mathbb{E}_ {\tilde{\mathbf{x}}_ t}[\mathbf{x}_ 0])$, we need to calculate the dynamic coefficient during training, which may be time-consuming.
>
> **Q2: re-ranking**
>
> The transformer generates with a beam size of 5.
> We do not use sampling because
> 1. **We strictly follow our baselines that all of them are using the beam search generation.**
> 2. **Diffusion LMs are not stable enough to keep achieving good results with simple sampling like the transformer**, as illustrated in Appendix I and our baselines. They have the potential to generate high quality contents, but random initialization will have a great impact on the generated results. Transformers do not face this problem of randomness.
>
> Take IWSLT14 as an example.
>
> |  | beam search | sampling |
> | --- | --- | --- |
> | transformer | 34.31 | 34.05 |
> | Ours Rerank | 35.02 | 33.30 |
>
> When we convert the beam search to sampling, there is a large drop for the reranked diffusion model.
> As illustrated in Table 1, we do not intend to claim that our method surpasses transformers, but rather to show that our method **has the potential to produce results comparable to transformers** in addition to outperforming existing diffusion models.
>
> **Q3: Table 3**
>
> We want to clarify that **the most closest and simplest baseline is the DDIM, not the Flow Matching.**
> Our method is a deterministic diffusion process or ODE with dicrete time step. We use the ODE framework to model the deterministic forward process (because DDPM or DDIM framework can not do this), so that our method is theoretically compatible with both the Diffusion process and Flow Matching. However, currently our method requires to predict $x_0$ in application and thus **reverses the diffusion process step by step**, which is actually a deterministic diffusion model, not a standard Flow Matching model.
> Therefore, the most closest and simplest baseline is the DDIM, not the Flow Matching. (The DDIM sampling of BitDiffusion is 11.37, slightly worse than Gaussian sampling)
>
> If we want to apply the flow matching to the same discrete embedding space, there are several problems:
> 1. If we predict $x_0$, the only difference to BitDiffusion is the optimal transport trajectory. As in the ablation study (Table 2) for language, this sometimes improves performance, but the effect is not large.
>
> Due to the high training cost, we currently only have results of 200K steps, where changing the trajectory to optimal transport does not make a significant difference.
> | | FiD|
> | --- | --- |
> | BitDiffusion repro | 22.12 |
> | BitDiffusion Flow | 21.05 |
> | Ours | 8.17 |
> | Ours Flow | 8.08 |
>
> 2. If we predict $\tilde{u}$, our method is currently not adaptable to this objective. As demonstrated in Q1-3.1, we have to solve the equation $\tau=\mathcal{T}(t,G(\mathbf{x}_0,\frac{\mathbf{x}-\mathbf{u}(\mathbf{x}_0,\tau)\mathbf{x}_0}{\mathbf{v}(\mathbf{x}_0,\tau)}))$ to get the $\tau$ with respect to the change of $\mathbf{x}$ in real time, which is currently inefficient to apply.

---

> ### Author Response · Authors · 2024-08-09
> **Response to Reviewer**
>
> We want to supplement that since solving the equation $\tau=\mathcal{T}(t,G(\mathbf{x}_0,\frac{\mathbf{x}-\mathbf{u}(\mathbf{x}_0,\tau)\mathbf{x}_0}{\mathbf{v}(\mathbf{x}_0,\tau)}))$ to get the $\tau$ with respect to the change of $\mathbf{x}$ in real time is difficult, this is why we use the equation 25 as an alternative. The equation 25 discretizes $t$ to finite steps and keeps track of previous $\tau$ to make it tractable. And therefore the discrete time step make our model the deterministic diffusion process rather than the ODE based flow matching with infinite continuous timesteps.
>
>
> In addition, if the reviewer still wants to know the performance of Flow Matching on binary coding, (although our method is not closely related to Flow Matching and is currently not efficient to the Neural ODE framework in application), we are training the model with the TorchCFM framework and will provide the results before the end of the discussion period.

---

> > ### Comment · Reviewer_1VFK · 2024-08-09
> > **Response to Authors**
> >
> > Thank you for the continued and detailed engagement with my questions.
> >
> > I appreciate this new analysis of the conditional vector field. As you have shown $\tilde{u}_t(\tilde{x}_t | \mathbb{E}[ x_0 | \tilde{x}_t])$ (the quantity you use during sampling) is an upper bound on $\mathbb{E} [ \tilde{u}_t (\tilde{x}_t | x_0) | \tilde{x}_t]$ (the quantity that you actually need to be using to be performing principled generative modelling). This is quite concerning since even if you train the network perfectly, you are not targeting the required quantity. I think this analysis should be included in the paper and it should be clearly stated that this departs from standard generative modelling objectives. You should also add discussion about why you still expect this to work, it is entirely not clear now what your framework is doing.
> >
> > I understand better now your experimental contributions and see that BitDiffusion is indeed very much like your method but without the boundary considerations. In light of your good experimental results, it seems that your method does have promise even if I don't believe right now it is well justified.
> >
> > I think the paper is quite borderline because of the unclear derivation of the methodology but good experimental results. I will raise my score to 5.

---

> ### Author Response · Authors · 2024-08-11
> **Response to Reviewer**
>
> We sincerely appreciate your response, and we are delighted to have addressed most of your concerns.
> We attach great importance to your suggestions and conduct some analytical experiments for better understanding.
>
> **Q: Why our objective works?**
>  1. In practice, errors come not only from the theory but also from the capability of neural networks. And **the error caused by optimizing the upper bound is much smaller than the error from the neural network.** Therefore, **the theoretical error caused by optimizing the upper bound** is negligible.
>
> The symbols in the formula may sometimes be confusing, and substituting them into numbers will lead to a more intuitive understanding. We add an additional analysis experiment on IWSLT14 to reveal the thinking behind our formula.
>
>
> | Objective | $\mathbb{E}_ {\tilde{\mathbf{x}}_ t}\Vert\mathbf{x}_ 0 - \mathbf{x}_ \theta\Vert$ | $\mathbb{E}_ {\tilde{\mathbf{x}}_ t}\Vert\tilde{u}_ t(\tilde{\mathbf{x}}_ t\vert\mathbf{x}_ 0) - \tilde{u}_ t(\tilde{\mathbf{x}}_ t\vert\mathbf{x}_ \theta)\Vert$ | $\mathbb{E}_ {\tilde{\mathbf{x}}_ t} [p(\mathbf{x}_ \theta \in C_{\mathbf{x}_0})] $ | BLEU (BLEU-1/2/3/4)|
> | --- | --- | --- | --- | --- |
> | $\Vert\mathbf{x}_ 0 - \mathbf{x}_ \theta\Vert$ |  8.4358 | 1.5613 | 51.81% | 33.42 (68.0/42.0/27.7/18.6) |
> | $\Vert\tilde{u}_ t(\tilde{\mathbf{x}}_ t\vert\mathbf{x}_ 0) - \tilde{u}_ t(\tilde{\mathbf{x}}_ t\vert\mathbf{x}_ \theta)\Vert$ |  8.4061 | 1.5518 | 52.34% | 33.49 (68.0/41.9/27.7/18.6) |
>
> We demonstrate the Expectation of $\mathbf{x}_ 0$'s and $\tilde{u}_ t$'s errors on the test set. It is easy to observe that, with the dynamic coeffficient $\frac{T-r\times G(\mathbf{x}_0, \boldsymbol{\epsilon})}{T}$, the value of $\mathbf{x}_0$'s error (8.4358) is much larger (than 1.5613). This supports Lines 180-182 that predicting $\tilde{v}_t^\theta$ will amplify the error on $\mathbf{x}_0$, whereas predicting $\mathbf{x}_0$ will reduce the error of neural networks on $\tilde{u}_t$. **Therefore, predicting $\mathbf{x}_0$ is beneficial to reducing the impact of the prediction error of the neural network compared with $\tilde{u}_t$**.
>
> In addition, we convert the objective to $\Vert\tilde{u}_ t(\tilde{\mathbf{x}}_ t\vert\mathbf{x}_ 0) - \tilde{u}_ t(\tilde{\mathbf{x}}_ t\vert\mathbf{x}_ \theta)\Vert$ where the neural network still predicts $\mathbf{x}_0$. We find that the error expectations of $\mathbf{x}_0$ and $\tilde{u}_t$ decrease 0.0297 (0.35%) and 0.0095 (0.61%), respectively. This means that **the error caused by optimizing the upper bound is basically negligible compared to the error of the neural network.** Furthermore, predicting the upper bound has almost no impact on the final performance (0.07 of the BLEU score).
>
> 2. **Discrete modeling may be more robust to neural network errors than the continuous modeling.**
>
> We compare the DDPM and BitDiffuison with different image embedding types over the eval set of cifar10.
> | Models | $\mathbb{E}_ {\tilde{\mathbf{x}}_ t} [p(\mathbf{x}_ \theta\in C_ {\mathbf{x}_ 0})]$ |
> | --- | --- |
> | DDPM | 1.18% |
> | BitDiffusion | 25.74% |
>
> For continuous embedding, each pixel is represented with a float number in [-1,1], where the predictions will be discretized to integers in [0,255]. It is really hard to recover the original pixels with only one step (1.18%). For binary coding, where the pixel representation is an 8-dimensional vector, it is more easy to recover $\mathbf{x}_0$ due to a larger absorption state (25.74%).
> **This means the discrete embedding space can be more robust to errors of neural networks.**
> (*It's worth noting that although discrete embedding is more robust, it weakens the continuity between adjacent pixels and currently cannot exceed the performance of continuous models.*)

---

> ### Author Response · Authors · 2024-08-11
> **Response to Reviewer**
>
> We would like to express our sincere gratitude to you again. Your suggestions are of great help in improving the quality of our paper.
>
> We believe the following minor revisions will improve the clarity of our paper.
> 1. In **Training Objective** part of Section 3.3, we will change the equation 24 to a quick introduction of the $\tilde{u}_ t$ and move the equation to this part.
>  Then we will add the derivation from $\Vert\tilde{u}_ t(\tilde{\mathbf{x}}_ t|\mathbf{x}_ 0)-\tilde{u}_ t(\tilde{\mathbf{x}}_ t|\mathbf{x}_ \theta)\Vert$ to $\Vert\mathbf{x}_ 0-\mathbf{x}_ \theta\Vert$ as discussed above, including the upper bound. Therefore, our method covers the entire framework of the Flow Matching in theory.
>
> 2. In **Reverse Process** part of Section 3.3, we will add an explaination that *solving the equation $\tau=\mathcal{T}(t,G(\mathbf{x}_0,\frac{\mathbf{x}-\mathbf{u}(\mathbf{x}_0,\tau)\mathbf{x}_0}{\mathbf{v}(\mathbf{x}_0,\tau)}))$ to get the $\tau$ with respect to the change of $\mathbf{x}$ in real time is difficult*. Therefore, the theory and practice are distinguished in this part, where we explain how to implement our method.
>
> 3. In Section 4, we will add analyses about the expectation of errors as demonstrated above.
>
> 4. Other derivations and discussions that do not affect reading will be added to the appendix.

---

> > ### Comment · Reviewer_1VFK · 2024-08-11
> > **Response to Authors**
> >
> > I appreciate the empirical analysis of the error induced when switching from the $u_t$ based objective to the $x_0$ based objective. I also think the paper's clarity will be improved greatly including this derivation of the loss in the main with its relation to flow matching, it will definitely help the reader's understanding with your method. I will raise my score to 6 due to my main concern of the unclear derivation of the method being mostly resolved.

---

> > > ### Author Response · Authors · 2024-08-12
> > > **Response to Reviewer**
> > >
> > > Thank you again for your patience and kindness, your suggestions and guidance have been very helpful.

---

### Official Review · Reviewer_uQEt · 2024-07-09

**Soundness:** 2
**Presentation:** 3
**Contribution:** 2
**Rating:** 5
**Confidence:** 4

**Summary:**

In this study, the authors propose a framework to address the discrepancy between discrete data and continuous modeling in diffusion processes. The approach involves a two-step forward process that first estimates the boundary as a prior distribution, then rescales the forward trajectory to build a boundary conditional diffusion model. The reverse process is proportionally adjusted to ensure the learned contours yield more accurate discrete data. Experimental results show the method's strong performance in language modeling and discrete image generation tasks, outperforming previous models in several areas and establishing a new state-of-the-art for categorical image generation on the Cifar-10 dataset.

**Strengths:**

- The authors have conducted comprehensive experiments, providing thorough and complete details in their exploration.
- The novelty of the study lies in its discussion of the concept of "discrete area", presenting a fresh perspective in the field.

**Weaknesses:**

- The explanation as to why this new framework can enhance performance is not clearly articulated in the study.
- From my understanding, Algorithm 1 appears to be equivalent to training original diffusion models with $t\sim U[t_0,T]$ as opposed to $t\sim U[1,T]$.

**Questions:**

- Is there a possibility that the boundaries of the discrete areas of two distinct data points might intersect? If so, why would pushing the noise to the boundary be beneficial? Could certain perturbations potentially lead to inaccuracies in the final result?
- According to Equation (14), shouldn't the input of $\Psi^{-1}([\cdot,\cdot])$ be the boundary point and the "crossing boundary time"? If so, why does line 8 of Algorithm 2 suggest that a sequence of $\tau$ ranging from $T$ to $r\times t_0$ can serve as the input of $\Psi^{-1}([\cdot,\cdot])$?
- also please refer to weaknesses.

If these questions are addressed, I would consider revising the score.

**Limitations:**

The authors do acknowledge the limitations of their work, which is commendable. Furthermore, I recommend conducting theoretical analysis, such as exploring the convergence properties of their new framework.

---

> ### Author Rebuttal · Authors · 2024-08-06
>
> We sincerely appreciate your meticulous review and the valuable comments. Please kindly find our response below.
>
> **Q1: Why can our framework enhance performance**
>
> 1) **From the perspective of Motivation**
>
> As illustrated in Introduction and Figure 1, we find that the continuous diffusion processes oversimplify the objective of discrete modeling. The discrete data are treated as single points, while they are actually areas in the diffusion space. This means diffusion models learned on the oversimplified objective generate the sub-optimal probability density contour, which is inconsistent with the discrete priors (Lines 38-41 and Figure 1A). Therefore, **our method takes the discrete prior into consideration and design a more appropriate diffusion trajectory for discrete modeling problems** (Lines 42-65 and Figure 1B).
>
> 2) **From the perspective of Methodology**
>
> We theoretically derive the diffusion process conditioned on the discrete priors and obtain the sampling distribution of the corresponding forward process $\tilde{p}_t(\mathbf{x}|\mathbf{x}_0)$ (Equations 21 and 22). It is easy to calculate that the $\tilde{p}_0(\mathbf{x}|\mathbf{x}_0)$ is exactly the discrete boundary and $\tilde{p}_T(\mathbf{x}|\mathbf{x}_0)$ is a Gaussian distribution. During the inference stage, i.e., reversing the time step from $T$ to $0$, the density probability will gradually be consistent with the contour when approaching the boundary, which is inline with Lines 46-51 and Figure 1B. Hence, **our method will precisely guide the random noise into the discrete area.** Figure 2 demonstrates sample points exactly calculated by our proposed method (Equation 22) and the probability contours calculated by our method can be easily adapted to different discrete boundaries and noising trajectories, which are inline with our expectations in Figure 1 of the Introduction Section.
>
> 3. **From the perspective of Experiment**
>
> Our framework is constructed on Difformer and BitDiffusion with the same configurations for both language and image generation tasks. As in Table 1 and 3, **our method demonstrates significant improvements over them**.
> Besides, ablations in table 4 reveal that, with the increase of the confidence factor $r$, i.e., increasing the influence of the discrete prior, the discrete modeling performance improves in all aspects.
>
> **Q2: Algorithm 1**
>
> You can consider $t\sim U[t_0,T]$. However, $t_0$ is not a constant but a random variable $t_0=G(\mathbf{x}_0,\boldsymbol{\epsilon})$ as illustrated in equation 12.
>
> **Sampling the $t$ from $U[1,T]$ and then transform it to $\tau$ with $\mathcal{T}(t,G(\mathbf{x}_0,\boldsymbol{\epsilon}))$ is equivalent to directly sampling $\tau$ from $U[G(\mathbf{x}_0,\boldsymbol{\epsilon}), T]$.** We choose the former procedure because it is easy to perform in parallel. Algorithm 1 is just a commonly used element substitution technique in sampling.
>
> **Q3: Boundaries of discrete areas**
>
> *Due to the character limit of the rebuttal, this is only an outline of our answer. Please refer to the comment for more details.*
>
> 1. **The discrete area of two distinct data points will not intersect.**
>
> As illustrated in Lines 34-37 and Equation 6, points in the discrete area will not be attributed to any other area by definition.
>
> 2. **Boundaries of two areas can overlap with the confidence factor $r=1$, but this is impossible when $r<1$.**
>
> When the confidence factor $r=1$, as illustrated in Equations 7 and 8, the boundary is where the probabilities of points belonging to the neighbour areas equal. When the confidence factor $r$ decreases, the range of each discrete area is gradually reduced and there is no overlap between boundaries.
>
> 3. **Pushing the noise to the boundary is beneficial.**
>
> - **Empiricaly**, previous work like Difformer and Dinoiser and our Table 1 have revealed that **increasing the minimal noise scale can benefit the performance** (Lines 217-219). And it is obvious that **pushing the noise to the boundary is precisely what increases the minimal noise scale**.
>
> - **Theoretically**, probability density inside the discrete area can be ignored.
> Based on the observation of the discrete area that any point inside this area will be attributed to the corresponding discrete data (Lines 34-37), we can easily get the derivation: **we don't need to model the probability density inside the discrete area because our diffusion model only needs to guide a noise to get into this area, where the accurate end position doesn't matter.** Therefore, pushing the noise to the boundary does not cause theoretical damage to the probability modeling. And the dffusion model can be less disrupted by invalid data, i.e., points inside the area, during the training stage.
>
>
> 4. **Certain perturbations can lead to inaccuracies in the final result without our confidence factor.**
>
> We use the confidence factor $r$ (Lines 137-147) to control the influence of perturbations. When we set a smaller $r$, the learned reverse process will be robust to perturbations.
>
> **Q4: Input of function $\Psi^{-1}([\cdot,\cdot])$**
>
> **The input of $\Psi^{-1}([\cdot,\cdot])$ is valid for any arbitrary pair of point $\tilde{\mathbf{x}}_t$ and time $\tau$.**
> Functions $\Psi$ and $\Psi^{-1}$ in Equation 14 are directly extended from Equations 4 and 5, which construct the invertible relationship between any point-time pair ($\tilde{\mathbf{x}}_t$-$\tau$) and their initial noise $\boldsymbol{\epsilon}$.
>
> In Equation 14 of Section 3.1, we take the boundary point $\mathbf{x}_{t_0}$ and the crossing boundary time $t_0$ pair as input because we just want to estimate this boundary. However, this does not mean the input of $\Psi^{-1}$ can only be the pair of boundary point and crossing boundary time, while any other point-time pairs are still valid. This is why we use $\Psi^{-1}$ to update the trajectory in Line 8 of Algorithm 2.

---

> > ### Comment · Reviewer_uQEt · 2024-08-11
> >
> > Thanks for your detailed reply.
> >
> > Now, I have no concerns on the motivation of this work. The motivation is reasonable.
> > But I still feel confused towards the design of the sampling (Algorithm 2), can you provide more illustration on this? It is beneficial to give more clear presentation of this.

---

> ### Author Response · Authors · 2024-08-06
> **Detailed answer for Q3: Boundaries of discrete areas**
>
> **Q3: Boundaries of discrete areas**
>
> 1. **The discrete area of two distinct data points will not intersect.**
>
> As illustrated in Lines 34-37 and Equation 6, points in the discrete area will not be attributed to any other area by definition.
>
> Take language modeling as an example, the discrete area of a token 'A' is the set of all points in the embedding space with the largest dot similarity to token 'A' than other tokens in the vocabulary. This means almost any point in the embedding space will be attributed to only one token. When the dot similarities of a point to different tokens equal, it means the point lies at the boundary of these discrete areas.
>
> Besides, the discrete area is convex. This means all points within the boundary will only belong to this area, i.e., we can safely discribe the area with boundaries as in Section 3.1. The quick proof of convex is:
> > Suppose the embedding of token 'A' is $\mathbf{x}_0$, given two points $\mathbf{y}^0$ and $\mathbf{y}^1$ in the embedding space.
> >
> > If $\mathbf{y}^0$ and $\mathbf{y}^1$ are inside the discrete area of token 'A' and for all embeddings $\mathbf{x}_i$ of other tokens,
> >
> > there is: $\mathbf{x}_0\cdot\mathbf{y}^0\geq\mathbf{x}_i\cdot \mathbf{y}^0$ and $\mathbf{x}_0\cdot\mathbf{y}^1\geq\mathbf{x}_i\cdot \mathbf{y}^1$.
> >
> > For any other points $\hat{\mathbf{y}} = t \ \mathbf{y}^0 + (1-t)\ \mathbf{y}^1, t\in[0,1]$,
> >
> > there is $\mathbf{x}_0\cdot\hat{\mathbf{y}} = t(\mathbf{x}_0\cdot\mathbf{y}^0)+(1-t)(\mathbf{x}_0\cdot\mathbf{y}^1)\geq \mathbf{x}_i\cdot\hat{\mathbf{y}}$.
> >
> > Therefore, the discrete area is convex.
>
> 2. **Boundaries of two areas can overlap with the confidence factor $r=1$, but this is impossible when $r<1$.**
>
> When the confidence factor $r=1$, as illustrated in Equations 7 and 8, the boundary is where the probabilities of points belonging to the neighbour areas equal.
> Consider a simplified situation of only two tokens 'A' and 'B', which is the binary classification, the boundary of token 'A' overlaps the boundary of token 'B', where $p(\mathbf{x} \in A)=p(\mathbf{x}\in B) = 0.5$.
>
> When the confidence factor $r$ decreases, the range of each discrete area is gradually reduced. For example, we may take $p(\mathbf{x} \in A)=0.7 > p(\mathbf{x}\in B)$ as the boundary of 'A' and $p(\mathbf{x} \in B)=0.7$ as the boundary of 'B'. In this case, $0.3<p(\mathbf{x} \in A)<0.7$ is an unattributed area and there is no overlap between the boundaries of 'A' and 'B'.
>
> 3. **Pushing the noise to the boundary is beneficial.**
>
> - **Empiricaly**, previous work like Difformer and Dinoiser and our Table 1 have revealed that **increasing the minimal noise scale can benefit the performance** (Lines 217-219). In both their experiments and our Table 1, comparing to the original Diffusion LMs like DiffuSeq and SeqDiffuSeq, increasing the minimal noise scale (difformer and dinoiser) can improve the performance. And it is obvious that **pushing the noise to the boundary is precisely what increases the minimal noise scale**.
>
> - **Theoretically**, probability density inside the discrete area can be ignored.
> > Let’s make a simple analogy. If we want to launch a satellite around the moon, we can treat the moon as a point mass. This is similar to the traditional continuous diffuison process. When it comes to landing on the moon, we have to take into account the shape of the moon, while the gravity field inside the moon does not matter. This is what our discrete problem looks like.
>
> Based on the observation of the discrete area that any point inside this area will be attributed to the corresponding token (Lines 34-37), we can easily get the derivation: **we don't need to model the probability density inside the discrete area because our diffusion model only needs to guide a noise to get into this area, where the accurate end position doesn't matter.** Therefore, pushing the noise to the boundary does not cause theoretical damage to the probability modeling. And the dffusion model can be less disrupted by invalid data, i.e., points inside the area, during the training stage.
>
> 4. **Certain perturbations can lead to inaccuracies in the final result without our confidence factor.**
>
> We use the confidence factor $r$ (Lines 137-147) to control the influence of perturbations. Given the above example of only two tokens 'A' and 'B'. When the confidence factor $r=1$, points $\mathbf{x}$ on the boundary of 'A' follow $p(\mathbf{x}\in A)=p(\mathbf{x} \in B) = 0.5$. Therefore, any perturbation to the prediction of the learned diffusion model may lead to a reverse of the attribution of $\mathbf{x}$. To alleviate this problem, we can decrease the factor $r$ so that points on the boundary have higher confidence, e.g., $p(\mathbf{x}\in A)=0.7>p(\mathbf{x}\in B)$, which is more robust to perturbations.
> In practice, $r=1$ works well for tasks with strong conditions and we should decrease $r$ for unconditional tasks.

---

> ### Author Response · Authors · 2024-08-11
> **Response to Reviewer**
>
> We sincerely appreciate your response, and we are delighted to have addressed some of your concerns.
>
> **Q: The design of sampling (Algorithm 2)**
>
> Lets start from the sampling process of DDIM under our symbol denotion.
>
> > 1. $\mathbf{x}_ T \sim \mathcal{N}(0,1)$
> > 2. for $t=T,\dots,1$ do
> > 3. `  `$\hat{\mathbf{x}}_ 0 = \mathbf{x}_\theta(\mathbf{x}_t, t)$ (*Pseudo Target*)
> > 4. `  `$\boldsymbol{\epsilon} = \frac{\mathbf{x}_t - \mathbf{u}_t\hat{\mathbf{x}}_0}{\mathbf{v}_t}$ (*Trajectory Alteration*)
> > 5. `  `$\mathbf{x}_ {t-1} = \mathbf{u}_ {t-1}\hat{\mathbf{x}}_ 0 + \mathbf{v}_ {t-1}\boldsymbol{\epsilon}$ (*Previous Sample*)
>
> This is a synchronous trajectory update where the algorithm directly updates the start point $\boldsymbol{\epsilon}$ of the trajctory based on the current prediction $\hat{\mathbf{x}}_0$.
>
> Our method requires one more step of timestep rescaling $\tau = \mathcal{T}(t, G(\hat{\mathbf{x}}_ 0,\boldsymbol{\epsilon}))$, which takes the current trajctory start point as the input. However, the trajectory alteration (step 4 above) also requires the current timestep $\tau$ as the input $\hat{\boldsymbol{\epsilon}} = \frac{\tilde{\mathbf{x}}_ t - \mathbf{u}_ \tau\hat{\mathbf{x}}_ 0}{\mathbf{v}_ \tau}$. Therefore, our procedure can not be synchronous and we must choose $\tau$ or $\hat{\boldsymbol{\epsilon}}$ to be asynchronous.
>
> As discussed with reviewer 1VFK (Q3.3 fixing the initial path), we find that even discarding the step of Trajectory Alteration leads to almost no performance degradation.
>
> (*1. From the perspective of uncertainty, if the learned diffusion model predicts a different $\mathbf{x}_0$, we have to update the original noise because currently we are in a different trajectory. If the learned model keeps changing the predicted target, which means it is uncertain about where to go, updating the trajectory correspond to an unreliable target may not be beneficial for the denoising process. If the learned model rarely changes its prediction, it means we are on the right path and no updates are needed. Therefore, it would be a viable option to discard this step of Trajectory Alteration to make decoding faster.
> 2. From the perspective of experiments, as illustrated in Table 3, if we train with the rescaled forward process and decode with unrescaled trajectory, there will be only a slight performance degradation. The performance improvement of our model mainly comes from the training process, and some small changes in the decoding process may not have much impact. In practical application, we just discard the trajectory update for simplicity in our experiments and find almost no performance degradation.*)
>
> Therefore, we choose $\hat{\boldsymbol{\epsilon}}$ to be asynchronous and the steps in the for loop (line 2) becomes:
> > 3. `  `$\hat{\mathbf{x}}_ 0 = \mathbf{x}_\theta(\tilde{\mathbf{x}}_t, t)$ (*Pseudo Target*)
> > 4. `  `$\tau = \mathcal{T}(t-1, G(\hat{\mathbf{x}}_ 0,\hat{\boldsymbol{\epsilon}}))$ (*Trajctory Rescaling*)
> > 5. `  `$\tilde{\mathbf{x}}_ {t-1}=\mathbf{u}_ \tau \hat{\mathbf{x}}_ 0 + \mathbf{v}_ \tau \hat{\boldsymbol{\epsilon}}$ (*Previous Sample*)
> > 6. `  `$\hat{\boldsymbol{\epsilon}} = \frac{\tilde{\mathbf{x}}_ {t-1} - \mathbf{u}_ \tau\hat{\mathbf{x}}_ 0}{\mathbf{v}_ \tau}$ (*Asynchronous Trajectory Alteration*)
>
> where we add the Trajctory Rescaling step and move the Asynchronous Trajectory Alteration to the end.
> Other steps in Algorithm 2 is to keep track of current $\tau$ and $\hat{\boldsymbol{\epsilon}}$.
>
> It's worth noting that we can not loop the $\tau$ as $t$ because $\tau$ is dynamic and different for each pair of $\mathbf{x}_0$ and $\boldsymbol{\epsilon}$. In the same iteration round, different $\mathbf{x}_0$ corresponds to different $\tau$.
>
> Therefore, Algorithm 2 can complete the iterative prediction of $\mathbf{x}_0$ with subtal errors. And in the analytical experiments discussed with the reviewer 1VFK, the errors in the sampling stage have almost no impact on the final results.
>
> (*There is a typo in Algorithm 2 that $\tilde{\mathbf{x}}_ t$ in Line 6 should be $\tilde{\mathbf{x}}_ {t-\Delta t}$*)

---

> ### Author Response · Authors · 2024-08-13
> **Response to Reviewer**
>
> We hope to supplement the above description in a more vivid way.
>
> The sampling process of DDIM can be demonstrated as:
> $$
> \begin{aligned}
> \mathbf{x}_ \theta ^{t+1} & \\\\
> \searrow &\\\\
> \boldsymbol{\epsilon} \rightarrow &\mathbf{x}_ t
> \end{aligned} \Rightarrow
> \begin{aligned}
> &\quad \mathbf{x}_ \theta ^ {t}\\\\
> &\nearrow \\\\
> \boldsymbol{\epsilon} \rightarrow &\mathbf{x}_ t
> \end{aligned} \Rightarrow
> \begin{aligned}
> &\qquad \ \mathbf{x}_ \theta ^ t\\\\
> & \qquad \ \downarrow \\\\
> \boldsymbol{\epsilon} \rightarrow &\mathbf{x}_ t \rightarrow \boldsymbol{\epsilon}'
> \end{aligned} \Rightarrow
> \begin{aligned}
> &\qquad \ \mathbf{x}_ \theta ^ t\\\\
> & \qquad \ \downarrow \quad \searrow \\\\
> \boldsymbol{\epsilon} \rightarrow &\mathbf{x}_ t \rightarrow \boldsymbol{\epsilon}' \rightarrow \mathbf{x}_ {t-1}
> \end{aligned}
> $$
>
> In this process, the reverse trajectory is $\boldsymbol{\epsilon} \rightarrow \mathbf{x}_ t \rightarrow \mathbf{x}_ 0$.
> When the prediction of $\mathbf{x}_ 0$ changes, the trajectory is updated with $\mathbf{x}_ t$ as the fixed point $\boldsymbol{\epsilon}' \rightarrow \mathbf{x}_ t \rightarrow \mathbf{x}_ \theta$, which can be demonstrated as:
>
> $$
> \begin{aligned}
> \boldsymbol{\epsilon} \rightarrow &\mathbf{x}_ t \rightarrow \mathbf{x}_ 0
> \end{aligned} \Rightarrow
> \begin{aligned}
> &\qquad\ \mathbf{x}_ \theta\\\\
> &\quad \nearrow \\\\
> \boldsymbol{\epsilon} \rightarrow &\mathbf{x}_ t \rightarrow \mathbf{x}_ 0 \\\\
> \nearrow&\\\\
> \boldsymbol{\epsilon}' \quad\ &
> \end{aligned} \Rightarrow
> \begin{aligned}
> &\qquad\qquad\ \mathbf{x}_ {\theta}\\\\
> &\qquad \quad \nearrow \\\\
> &\qquad\ \mathbf{x}_ {t-1}\\\\
> &\quad \nearrow \\\\
> \boldsymbol{\epsilon} \rightarrow &\mathbf{x}_ t \rightarrow \mathbf{x}_ 0 \\\\
> \nearrow&\\\\
> \boldsymbol{\epsilon}' \quad\ &
> \end{aligned}
> $$
> Therefore, DDIM algorithm can be simplified as:
> > 1. $\mathbf{x}_ T \sim \mathcal{N}(0,1)$
> > 2. for $t=T,\dots,1$ do
> > 3. `  `$\hat{\mathbf{x}}_ 0 = \mathbf{x}_\theta(\mathbf{x}_t, t)$ (*Pseudo Target*)
> > 4. `  `$\mathbf{x}_ {t-1} = \mathbf{u}_ {t-1}\hat{\mathbf{x}}_ 0 + \mathbf{v}_ {t-1}\frac{\mathbf{x}_t - \mathbf{u}_t\hat{\mathbf{x}}_0}{\mathbf{v}_t}$ (*Previous Sample*)
>
>
> In our algorithm 2, the timestep rescaling $\tau = \mathcal{T}(t, G(\mathbf{x}_ 0, \boldsymbol{\epsilon}))$ takes both $\mathbf{x}_ 0$ and $\boldsymbol{\epsilon}$ as the input. **This means errors in the predicted $\mathbf{x}_ \theta$ will be maginified if we update $\boldsymbol{\epsilon}$ before rescaling the timestep.** Therefore, there will be two different solutions:
> 1. One is our implemention, where we directly discard the Trajectory Alteration step, and the fixed point of our reverse trajectory is $\boldsymbol{\epsilon}$:
> $$
> \begin{aligned}
> \boldsymbol{\epsilon} \rightarrow &\tilde{\mathbf{x}}_ t \rightarrow \mathbf{x}_ 0
> \end{aligned} \Rightarrow
> \begin{aligned}
> &\qquad \qquad \mathbf{x}_ \theta\\\\
> &\qquad \ \ \ \nearrow\\\\
> &\qquad \tilde{\mathbf{x}}_ t' \\\\
> &\ \ \nearrow \\\\
> &\boldsymbol{\epsilon} \rightarrow \tilde{\mathbf{x}}_ t \rightarrow \mathbf{x}_ 0 \\\\
> \end{aligned}\Rightarrow
> \begin{aligned}
> &\qquad\qquad\qquad \mathbf{x}_ \theta\\\\
> &\qquad\qquad \ \ \ \nearrow\\\\
> &\qquad \qquad \tilde{\mathbf{x}}_ {t-1}\\\\
> &\qquad \ \ \ \nearrow\\\\
> &\qquad \tilde{\mathbf{x}}_ t' \\\\
> &\ \ \nearrow \\\\
> &\boldsymbol{\epsilon} \rightarrow \tilde{\mathbf{x}}_ t \rightarrow \mathbf{x}_ 0 \\\\
> \end{aligned}
> $$
>
> It is worth noting that, in this situation, the Trajectory Alteration step is just like a place holder and does not change the value of $\boldsymbol{\epsilon}$, because the Previous Sample (Step 5) and Asynchronous Trajectory Alteration (Step 6) are exaclty the same equation.
>
> 2. The Asynchronous Trajectory Alteration works on the other solution where we still take $\tilde{\mathbf{x}}_ t$ as the fixed point. In this situation, the Previous Sample step is $\tilde{\mathbf{x}}_ {t-1} = \mathbf{u}_ \tau\hat{\mathbf{x}}_ 0 + \mathbf{v}_ \tau \frac{\tilde{\mathbf{x}}_ t-\mathbf{u}_ {\tau_ \text{prev}}\hat{\mathbf{x}}_ 0}{\mathbf{v}_ {\tau_ \text{prev}}}$.
> This means we calculate the $\tau$ with $\boldsymbol{\epsilon}$ but generate $\tilde{\mathbf{x}}_ {t-1}$ with the updated $\boldsymbol{\epsilon}'$. The corresponding reverse trajectory is:
> $$
> \begin{aligned}
> \boldsymbol{\epsilon} \rightarrow &\tilde{\mathbf{x}}_ t \rightarrow \mathbf{x}_ 0
> \end{aligned} \Rightarrow
> \begin{aligned}
> &\qquad\ \mathbf{x}_ \theta\\\\
> &\quad \nearrow \\\\
> \boldsymbol{\epsilon} \rightarrow &\tilde{\mathbf{x}}_ t \rightarrow \mathbf{x}_ 0 \\\\
> \nearrow&\\\\
> \boldsymbol{\epsilon}' \quad\ &
> \end{aligned} \Rightarrow
> \begin{aligned}
> &\qquad\qquad\ \mathbf{x}_ {\theta}\\\\
> &\qquad \quad \nearrow \\\\
> \boldsymbol{\epsilon} &\rightarrow \ \tilde{\mathbf{x}}_ {t-1}\\\\
> &\quad \nearrow \\\\
> \boldsymbol{\epsilon} \rightarrow &\tilde{\mathbf{x}}_ t \rightarrow \mathbf{x}_ 0 \\\\
> \nearrow&\\\\
> \boldsymbol{\epsilon}' \quad\ &
> \end{aligned}
> $$

---

> > ### Author Response · Authors · 2024-08-13
> > **Response to Reviewer**
> >
> > As mentioned in Lines 195-196 of our paper and discussion with reviewer 1VFK, there is almost no performance gap between the  two different fixed point solutions of our algorithm. Therefore, we choose the former one which is simple and stable. Besides, the Asynchronous Trajectory Alteration step (Line 8 in Algorithm 2) is optional in practice.

---

> > > ### Comment · Reviewer_uQEt · 2024-08-13
> > >
> > > Thanks for your detailed reply.
> > >
> > > I will raise the score to 5. More illustration on the algorithm should be added in revision.

---

> ### Author Response · Authors · 2024-08-13
> **Response to Reviewer**
>
> Thank you for your patience and kindness, your suggestions and guidance have been very helpful.
>
> We will make the following minor revisions to make it clearer.
> 1. Replace the current $\Delta t$ iteration into an easier notation.
> 2. Add the explaination about the Asynchronous Conflict between $\tau$ and $\epsilon$ and trajectory alteration strategy as discussed above.
> 3. Add comparison with DDIM in the appendix for better understanding.
> 4. Other derivations and discussions that do not affect reading will be added to the appendix.
>
> We believe these revisions can improve the clarity of our algorithm and we would like to express our sincere gratitude to you again.
>
> Besides, the asynchronous conflict problem is the same as the problem of $\tau=\mathcal{T}(t,G(\mathbf{x}_0,\frac{\mathbf{x}-\mathbf{u}(\mathbf{x}_0,\tau)\mathbf{x}_0}{\mathbf{v}(\mathbf{x}_0,\tau)}))$ discussed with reviewer 1VFK.
> > In **Reverse Process** part of Section 3.3, we will add an explaination that *solving the equation $\tau=\mathcal{T}(t,G(\mathbf{x}_0,\frac{\mathbf{x}-\mathbf{u}(\mathbf{x}_0,\tau)\mathbf{x}_0}{\mathbf{v}(\mathbf{x}_0,\tau)}))$ to get the $\tau$ with respect to the change of $\mathbf{x}$ in real time is difficult*. Therefore, the theory and practice are distinguished in this part, where we explain how to implement our method.
>
> Therefore, we will combine these two explanations.

---

### Official Review · Reviewer_m5hg · 2024-07-13

**Soundness:** 3
**Presentation:** 3
**Contribution:** 3
**Rating:** 6
**Confidence:** 3

**Summary:**

A discrepancy exists when using continuous diffusion models to model discrete data, which has not been sufficiently addressed in past methods. This paper redesigns the forward and backward processes of the diffusion model to eliminate this issue. Experiments on translation tasks and CIFAR-10 effectively validate the proposed method's effectiveness.

**Strengths:**

1. The writing is clear and concise.

2. This paper aims to address an important and interesting question.

3. Experiments demonstrate that the proposed method surpasses all baselines, effectively validating its effectiveness.

**Weaknesses:**

Please see the Questions.

**Questions:**

1. In Sec. 4, the authors present an untrainable $p(W|x_0)$ given the embedding layer. If we employ a trainable $p_{\theta}(W|x_0)$ (e.g., Diffusion-LM [1]), will the discrepancy between discrete data and continuous modeling still exist? If it does, will your proposed method still bring improvements in this scenario?

2. This paper presents the results of translation tasks. I am curious to know whether the proposed method would be effective for models like Plaid [2], which can compute text likelihood. The discrepancy between discrete data and continuous modeling mainly occurs during the step of decoding the embedding into text, while the likelihood calculation does not require this decoding step.

3. In Table 3, the authors report FIDs of 51.27 and 30.97 for D3PM. However, the best generative FID on CIFAR-10 in the original D3PM paper is 7.34. Why didn't the authors report this result as a baseline? This requires further explanation.

[1] Diffusion-LM Improves Controllable Text Generation

[2] Likelihood-Based Diffusion Language Models

**Limitations:**

yes

---

> ### Author Rebuttal · Authors · 2024-08-06
>
> We sincerely appreciate your meticulous review and the valuable comments. Please kindly find our response below.
>
> **Q1: The embedding layer**
>
> 1) We want to clarify that the embedding layer we use is actually trainable, as illustrated in Lines 200 and 208. The $\theta$ is only used to denote the diffusion parameters in our paper.
>
> 2) The discrepancy between discrete data and continuous modeling exists for both fixed and trainable embedding layers.
>
> 3) Besides, we are fully aware that it has been demonstrated in Diffusion-LM that a trainable embedding layer achieves better performance than the fixed one.
>
> **Q2: Application scenarios of our framework**
>
> 1) There seems to be an embedding layer in Plaid as well (Section 3.1 in Plaid), so our framework is theoretically applicable. Likelihood calculation is also a similarity function that has the same effect as the dot product in our work, which is compatible with the $f(\mathbf{x},\mathcal{I})$ framework we defined.
>
> 2) Besides, as illustrated in Lines 28-30, our framework is specifically designed for continuous diffuison models, where the discrete diffusion process with the discrete state space will not face the problem in Figure 1A and Lines 38-41. Therefore, our framework will extend the generality of the widely used continuous diffusion processes, but does no help for discrete diffusion processes.
>
> **Q3: FiD of D3PM**
>
> We want to clarify that we have reported the original D3PM in Table 3 with the FiD of 7.34, first line (D3PM GAUSS) in the part of Discrete Ordinal Pixels.
>
> As demonstrated in Table 3 and Section 5, we use different type of image embeddings. Different embedding type possesses different level of continuous information. The D3PMs of 51.27 and 30.97 FiDs are the UNIFORM and ABSORBING versions with Categorical Pixels.

---

> > ### Comment · Reviewer_m5hg · 2024-08-11
> >
> > Thank you for your response. I've decided to maintain my score.

---

> > > ### Author Response · Authors · 2024-08-12
> > > **Response to Reviewer**
> > >
> > > Thank you for your response and kindness, your suggestions have been very helpful.

---

### Official Review · Reviewer_ooCU · 2024-07-18

**Soundness:** 3
**Presentation:** 2
**Contribution:** 3
**Rating:** 5
**Confidence:** 4

**Summary:**

The paper discusses a novel approach to discrete modeling using boundary conditions and diffusion processes. The primary contributions include:
- Development of a framework for discrete image generation that incorporates binary coding and pixel embedding.
- Introduction of an intermediate state to illustrate the correlation between discreteness and modeling difficulty.
- Extensive experimental evaluation on datasets like CIFAR-10, demonstrating competitive results compared to state-of-the-art models.
- Proposal of a new method for stochastic processes that improve model performance, especially in cases of categorical pixels.

**Strengths:**

- Originality: The paper introduces a unique method for addressing discrete modeling through a combination of binary coding and pixel embedding, which is innovative in the context of diffusion models.
- Quality: The experimental setup is robust, with detailed evaluation metrics and comparisons to existing methods. The results show significant improvements, indicating the effectiveness of the proposed approach.
- Clarity: The paper is well-structured, with clear explanations of the methodology and the underlying mathematical formulations. Figures and tables are used effectively to illustrate the results.
- Significance: The approach has broad applicability in various fields requiring discrete data generation, such as image and language modeling. The improvements in FID scores and other metrics highlight the potential impact of this work on the research community.

**Weaknesses:**

1. The approach is primarily tested on specific datasets like CIFAR-10. Expanding the evaluation to a wider range of datasets on high-resolution images could provide a more comprehensive assessment of the model's generalizability.
2. Language model experiments are mainly on translation tasks. Could you provide experiments on language modeling tasks, which is more common for benchmarking diffusion LMs.

**Questions:**

See weakness.

**Limitations:**

Limitation is mentioned in one sentence in the conclusion section, but not comprehensively discussed.

---

> ### Author Rebuttal · Authors · 2024-08-06
>
> We sincerely appreciate your meticulous review and the valuable comments. Please kindly find our response below.
>
> **Q: Expanding datasets for images and languages**
>
> 1) For both language and image tasks, our framework is constructed on our baselines, Difformer and BitDiffusion, with the same configurations. We strictly follow their benchmarks and datasets to ensure a fair comparison.
>
> 2) Our experiments across languages and images may have reflected the generalizability of our framework.
>
> 3) Since training the image model takes a week and training the language model takes at least two days (lines 719-720), we do not have enough computing resources to scale to a larger and more complex benchmark.
> We sincerely accept your suggestions and will try to expand our method to larger benchmarks when we have enough computing resources.
>
> 4) We try to conduct the experiment on ImageNet 64, but our computing resources are currently insufficient to complete it. We show the temporary results of the first 100K steps as follows.
> | Binary Coding | FiD |
> |  ----  | ----  |
> | BitDiffusion  | 47.82 |
> | Ours  | 31.68 |

---

### Author Response · Authors · 2024-08-14
**General Response to All Reviewers and ACs**

We sincerely thank the reviewers for their efforts and suggestions.

We would like to briefly summarize our discussions with the reviewers.

1. All reviewers agree that our method is novel and meaningful, which incorporates discrete areas as priors when modeling discrete problems with continuous diffusion processes.
2. All reviewers agree that our experiments are effective with comprehensive baselines and promising performances.
3. Reviewers are mainly confused about the equations and algorithms in our method, and we provide detailed derivations and explanations in the discussion and have addressed their concerns.
4. Reviewer 1VFK raises questions about why our theoretical design is effective, and we supplement additional analytical experiments and theoretical explanations to address these concerns.

According to the suggestions of reviewers uQEt  and 1VFK, we will make the following minor revisions to improve the clarity of our paper:

1. In **Training Objective** part of Section 3.3, we will change the equation 24 to a quick introduction of the $\tilde{u}_ t$ and move the equation to this part.
 Then we will add the derivation from $\Vert\tilde{u}_ t(\tilde{\mathbf{x}}_ t|\mathbf{x}_ 0)-\tilde{u}_ t(\tilde{\mathbf{x}}_ t|\mathbf{x}_ \theta)\Vert$ to $\Vert\mathbf{x}_ 0-\mathbf{x}_ \theta\Vert$ as discussed, including the upper bound. Therefore, our method covers the entire framework of the Flow Matching in theory.

2. In **Reverse Process** part of Section 3.3, we will add an explaination that *solving the equation $\tau=\mathcal{T}(t,G(\mathbf{x}_0,\frac{\mathbf{x}-\mathbf{u}(\mathbf{x}_0,\tau)\mathbf{x}_0}{\mathbf{v}(\mathbf{x}_0,\tau)}))$ to get the $\tau$ with respect to the change of $\mathbf{x}$ in real time is difficult*. Therefore, the theory and practice are distinguished in this part, where we explain how to implement our method.

3. In **Algorithm 2** part of Section 3.3, we will add an explaination about the asynchronous conflict between $\tau$ and $\epsilon$ (*which will be connected to the above explaination of $\tau=\mathcal{T}(t,G(\mathbf{x}_0,\frac{\mathbf{x}-\mathbf{u}(\mathbf{x}_0,\tau)\mathbf{x}_0}{\mathbf{v}(\mathbf{x}_0,\tau)}))$*) and trajectory alteration strategy as discussed. Besides, we will replace the notations of Algorithm 2 to easier ones.

4. In Section 4, we will add analyses about the expectation of errors as demonstrated in the response to reviewer 1VFK.

5. Other derivations and discussions that do not affect reading will be added to the appendix.

We again express our gratitude to the ACs and reviewers for their efforts.

---

### Decision · Program_Chairs · 2024-09-25

**Decision:**

Accept (poster)

**Comment:**

The authors present a model that adapts continuous diffusion processes to discrete modeling. The proposed method is focused on injecting guidance for discrete boundaries. The approach is validated on vision and language tasks.

The reviewers agreed on the novelty of the presented approach compared to reference methods. Two of them raised some theoretical concerns, but the authors addressed them in detail in the rebuttal stage. There are also some concerns regarding the experimental part of the work, especially for images where Cifar10 is used. However, the results for language applications and the novelty of the work are satisfactory enough, and the paper should be accepted.